# Photodegradation Process of Organic Dyes in the Presence of a Manganese-Doped Zinc Sulfide Nanowire Photocatalyst

**DOI:** 10.3390/ma14195840

**Published:** 2021-10-06

**Authors:** Adam Żaba, Svitlana Sovinska, Tetiana Kirish, Adam Węgrzynowicz, Katarzyna Matras-Postołek

**Affiliations:** Faculty of Chemical Engineering and Technology, Cracow University of Technology, Warszawska 24, 31-155 Cracow, Poland; adam.p.zaba@gmail.com (A.Ż.); svitlanastorm@mail.ru (S.S.); tanyakirish@gmail.com (T.K.); adam.wegrzynowicz@pk.edu.pl (A.W.)

**Keywords:** organic dyes, photocatalysis, ZnS, ligands, doping, nanowires

## Abstract

Zinc sulfide (ZnS) nanowires represent a promising candidate in many fields, including optoelectronics and photocatalysis because of their advantages such as excellent optical properties, chemical stability and an easy-scalable simple synthesis method. In this study, an energy-friendly microwave radiation process was used to develop the single-step, solvothermal process for the growth of manganese-doped zinc sulfide (ZnS) and undoped nanocrystals (NCs) in the forms of nanowires using two short amines as a stabilizer, e.g. ethylenediamine and hydrazine, respectively. ZnS nanowires doped with Mn atoms show absorbance in UV and in the visible region of the spectrum. The photocatalytic degradation of rhodamine B in the presence of Mn-doped and undoped ZnS nanocrystals illuminated with only a 6-W UV lamp has been comprehensively studied. The effect of Mn doping and the presence of a nanocrystal stabilizer on the degradation process was determined. It was found that the efficiency of a photocatalytic degradation process was strongly affected by both factors: the doping process of nanowires with Mn^2+^ atoms and the attachment of ligands to the nanocrystal surface.

## 1. Introduction

One-dimensional semiconducting nanomaterials, like nanorods or nanowires, have attracted much research interest in the last decade owing to their unique properties [1]. In particular, zinc sulfide (ZnS) nanowires or nanorods with good optoelectronic properties are made of low-cost and abundant materials. They have been intensively investigated because of their tunable optical and electrical properties and chemical stability [2]. For example, it has been demonstrated that ZnS nanowires are promising candidates for diverse applications, including photovoltaic devices [3], sensors [4], lasers and light emitting devices [5]. In this context, the controlled synthesis of ZnS nanowires/nanorods is essential for both fundamental and applied studies. 

In recent years, ZnS was also extensively studied as a photocatalytic material due to its excellent physicochemical properties as well as its stability, nontoxicity and abundance. However, the large band gap (3.7 eV) can slightly limit its utilization in photocatalysis. To overcome this challenge, great effort has been devoted to engineer the ZnS band gap to achieve high photoactivity under visible light irradiation mainly by doping and size-shape controlling processes [6]. The introduction of dopants, e.g. copper [7,8,9,10], nickel [10,11] or manganese [10], is one of the most intensively investigated strategies to enhance (with varying effects) the NC absorbance into visible light utilization. The second strategy to improve the photocatalytic activity of nanomaterials can be the preparation of nanomaterials in one-dimensional (1D), two-dimensional (2D) or three-dimensional (3D) forms with relatively large surface area. 

Recently, intrinsic forms of ZnS such as ZnS nanoribbons [12], quantum dots [13,14,15] or 3D nanospheres [16,17,18,19,20,21,22,23,24] have been utilized as photocatalysts in the degradation of water pollutants. The most popular shapes of ZnS NCs studied so far in these photodegradation processes are quantum dot nanoparticles and nanospheres [13,14,15,16,17,18,19,20,21,22,23,24]. For example, Lixiong Yin et al. [21] synthesized ZnS nanospheres using a facile homogeneous precipitation process. The size of the obtained materials depended on the pH of the solution. By increasing the pH, the diameter of the nanospheres decreases, but the photocatalytic properties increase. Ma et al. [24] showed that the use of different stabilizers in the synthesis influences the surface roughness of ZnS nanospheres. The spheres obtained with thiourea had a smooth surface, as opposed to the ones synthesized with thioacetamide. However, the roughness of the spheres does not significantly influence the degradation of organic dyes.

To the best of our knowledge, there are not many works on controlling optical and morphological structure properties simultaneously for doped and undoped ZnS one-dimensional NCs with a high surface area and tailored band gap for the photocatalytic degradation of water pollutants. Moreover, there is no information on the effect of the amine ligands used for the synthesis of NCs on their photocatalytic activity. 

Taking into account the information above, in this paper we report a facile microwave-assisted solvothermal approach to produce highly crystalline and photoactive one-dimensional ZnS and ZnS:Mn nanowires/nanorods with a high surface area (~200 m^2^/g) and band gap tailored using Mn doping and two commonly used short amine solvents— ethylenediamine and hydrazine—as a ligand. Due to the doping of Mn^2+^ ions into ZnS crystals, the nanomaterials show improved absorption in the visible range of the light spectrum. The effect of microwave and conventional heating on the structure of ZnS NCs is also presented. It is known that that microwave-assisted reaction have remarkable advantages in comparison with the conventional heating methods, e.g. rapid volumetric heating, higher selectivity and chemical reaction rate, shorter reaction time and higher product yield, which make this procedure more effective and cheaper from the economic point of view [25].

The as-prepared ZnS and ZnS:Mn NCs show photocatalytic activity in the process of degradation of rhodamine B (RhB). To the best of our knowledge, this is the first demonstration of a microwave-assisted synthesis of ZnS and ZnS:Mn nanowires with photoactive properties using ethylenediamine and hydrazine, respectively. The present work shows that the microwave-assisted solvothermal method is rapid and can be easily scaled-up in practical production in the future.

## 2. Materials and Methods

### 2.1. Materials

As precursors of Zn, S and Mn zinc nitrate hexahydrate (Zn(NO_3_)_2_٠6H_2_O) (>99%), thiourea (CH_4_N_2_S) (99%) and manganese acetate tetrahydrate (Mn(CH_3_COO)_2_٠4H_2_O) (>99%) used were from Sigma-Aldrich (Poznan, Poland). As solvents and, at the same time, stabilizers, we used two commonly applied amines, i.e. ethylenediamine (C_2_H_8_N_2_) (>99%, Sigma-Aldrich) and hydrazine monohydrate (N_2_H_4_) (98%, Sigma-Aldrich). For the photocatalytic tests, rhodamine B (RhB) (99%, Sigma-Aldrich, CAS no. 81-88-9) and P25 TiO_2_ (≥99.5%, Sigma-Aldrich) were utilized. All chemicals were used as received without further purification. 

### 2.2. Preparation of ZnS and ZnS:Mn Nanowires in MW

The procedures to prepare undoped ZnS nanowires stabilized with ethylenediamine (EN) and hydrazine (N_2_H_4_), respectively, were mostly the same as that used for doped ZnS:Mn nanowires. The main difference was the addition of manganese acetate as a Mn atom precursor to the reaction mixture. All ZnS NC syntheses were carried out in microwave (MW) reactor Magnum II (ERTEC, Wroclaw, Poland) in a PTFE vessel (the maximum capacity of the vessel was 108 cm^3^) with a maximum applied power of 600 W and frequency of 2.45 GHz (corresponding to a wavelength of 12.25 cm). The details concerning the amount of the used substrate and preparation conditions were listed in Table 1. Typically, a PTFE flask contains an aqueous solution of hydrazine monohydrate (V_N2H4_: V_H20_ = 2:1) in the case of ZnS and ZnS:Mn NCs stabilized with N_2_H_4,_ or an aqueous solution of ethylenediamine (V_EN_: V_H20_ = 2:1) in the case of ZnS and ZnS:Mn NCs stabilized with EN, zinc nitrate hexahydrate (2 mmol), thiourea (4 mmol) and manganese (II) acetate tetrahydrate (0.02 mmol) were added sequentially at a time interval of 30 s under intense stirring. The mixture was stirred at room temperature for the next 10 min. Subsequently, the stirring was stopped and the PTFE vessel was placed into the MW reactor and heated at 160–170 °C temperature for 20 min at 8 bar. Next, the reaction mixture was cooled to room temperature. After cooling, the as-synthesized ZnS and ZnS:Mn nanowires were collected by centrifugation and washed with methanol three times. Then, the NCs were dried at 60 °C in the oven for 12 h. The ZnS nanowires were obtained as above without manganese acetate. For a typical reaction procedure, about 0.6 g of dry nanopowder was received. Plain ZnS nanowires were obtained as above without adding the Mn atom precursor.

### 2.3. Preparation of ZnS Nanocrystals in Conventional Heating

A conventional synthesis of the ZnS:Mn NCs was conducted with a similar procedure as reported in Section 2.2. In a typical synthesis, zinc nitrate hexahydrate (2 mmol), thiourea (4 mmol) and manganese (II) acetate tetrahydrate (0.02 mmol) were added into an aqueous solution of ethylenediamine or hydrazine (V_EN/NH_: V_H20_ = 2:1). The mixture of the precursor solution was loaded into a safe glass pressure vessel (Q-Tube-35 ml pressure reactor) and heated in a aluminum block equipment with thermostat and a magnetic stirrer. The experiments were carried out in a pressure reactor with vigorous mixing and at a temperature of 170 °C for 20 min. After slow cooling to room temperature, the reaction solution was centrifuged, washed twice with distilled water and dried in a vacuum oven at 60 °C for 12 h.

### 2.4. Characterization of Nanowires

The morphology of the NCs was observed by transmission electron microscopy (TEM, FEI Tecnai Osiris S/TEM, Munich, Germany) using an acceleration voltage of 200 kV) with energy-dispersive X-ray spectroscopy (EDX) and high-angle annular dark-field scanning transmission electron microscopy (HAADF-STEM) analysis. The phase composition of the samples was tested with an X-ray diffraction (XRD) meter (Bruker D8-Advance, Karlsruhe, Germany). Diffuse reflectance ultraviolet-visible spectroscopy (DR UV-Vis) absorption spectra were recorded with a UV-2700 spectrophotometer (Shimadzu, Krakow, Poland) equipped with an ISR-2600 Plus Integrating Sphere Attachment. Photoluminescence spectroscopy (PL) was used to determine the emission and excitation spectra using a Shimadzu RF-6000 spectrofluorophotometer. The Mn doping atom concentration in ZnS nanocrystals was determined with microwave plasma atomic emission spectrometry (MP-AES) (model 420, Aligent Technologies, Warsaw, Poland ). Fourier-transform infrared spectroscopy (FT-IR) spectroscopy (6700 FT-IR, Nicolet, Warsaw, Poland) was used to study the presence of ligands on the surface of the NCs. N_2_ absorption/desorption isotherms and Brunauer–Emmett–Teller (BET) specific surface area were obtained with a multi-function adsorption instrument (ASAP 2020, Micromeritics, Bruxelles Belgium). Specific surface area were calculated by applying Brunauer–Emmett–Teller (BET) formalism. Additionally recorded adsorption-desorption isotherms were used to estimate the average pore diameter. Before the measurement, the samples were degassed under vacuum at 350 °C for 24 h to ensure complete removal of impurities from the surface of the sample. Subsequently the sorption measurement were conducted at −196 °C in liquid nitrogen. The photoelectrochemical characterization and was performed using a photoelectric spectrometer (Instytut Fotonowy, Krakow, Poland) composed of a stabilized 150 W xenon arc lamp, monochromator, and coupled with a P-IF 4.0 potentiostat. The working electrodes for photoelectrochemical experiments were prepared by the deposition of the ZnS NCs onto ITO coated polyethylene terephthalate film. The electrolyte solution was 0.1 M KNO_3_. Platinum and Ag/AgCl were used as auxiliary and reference electrodes, respectively. The electrodes were contacted and then pressed against an O-ring of a photoelectrochemical cell, the irradiated sample area was of 0.28 cm^2^. The spectral range from 200 nm to 700 nm (25 nm step). The IPCE values for different wavelengths were calculated.

### 2.5. Photocatalytic Activity Test

The photocatalytic performances of the samples were tested by degrading rhodamine B (RhB) solutions. An initial RhB solution (5 mg/L) was first prepared. Next, 0.01 g of the sample was added into ∼30 mL of RhB solution. The concentration of ZnS based NCs vs dye RhB was 67:1 by weight (typical 10 mg of ZnS NCs vs 0.15 mg RhB) and was constant for all photocatalytic tests. The suspension was magnetically stirred for half an hour to achieve an adsorption–desorption balance in dark conditions. The degradation process was then performed under the UV light illumination at 254 nm. The photocatalytic activity was assessed by degrading RhB under 6 W UV irradiation spectroline lamp (Model ENF-260C/FE, 50 Hz, 0.17 AMPS, 254 nm). Two milliliters of the solution were taken out at regular intervals. The supernatant was then isolated from the solution by centrifugation. The concentration of the supernatant was obtained through measuring the absorbance at 550 nm, which was characterized with the UV-vis spectrometer (StellarNet Incorporation, Warsaw, Poland). The degradation process of RhB was studied by monitoring the changes in the maximal absorbance at 550 nm. 

## 3. Results and Discussion

### 3.1. Characterization of the ZnS and ZnS:Mn Nanowires

The as-prepared ZnS NCs were systematically studied in terms of morphology, structural and optical properties using TEM with EDX, XRD, DR UV-Vis, FT-IR, PL spectroscopy, and BET surface area analysis. Table 1 presents the detailed experimental parameters for the one-step synthesis of ZnS NCs in microwave reactor carried out in a short time (20 min).

We studied the role of both parameters like Mn doping concentration in ZnS nanowires and the surface effect on ZnS NCs on the photocatalytic activity of nanomaterials in the degradation of RhB. Therefore, two amine solvents, i.e. ethylenediamine (NH_2_-CH_2_-CH_2_-NH_2_) and hydrazine (NH_2_-NH_2_), commonly used in the solvothermal process, were used for the formation of ZnS nanowires, respectively (Table 1). Additionally, in order to compare the MW effect on the morphology of ZnS NCs, and, consequently, on the photocatalytic activity of the NCs, the synthesis under conventional heating in similar conditions was studied for two selected samples of ZnS:Mn with composition showing the highest photocatalytic activity in the degradation of RhB.

The phases and the purities of the nanomaterials were investigated with the XRD analysis. Figure 1 indicate XRD patterns of ZnS and ZnS:Mn NCs samples prepared with different content of manganese and ethylenediamine as a stabilizer (Figure 1a); with different types of using the heat source (Figure 2b) and with a different stabilizer (Figure 2c). The strong and broad diffraction peaks appearing in the XRD diagrams at 27.0°, 28.6°, 30.5°, 39.6°, 47.9°, 52.0° and 56.6° have obvious relevance to the well-crystallized hexagonal wurtzite ZnS structures (W, JCPDS#96-110-0045) matching relatively well with the crystal planes of (100), (002), (101), (102), (110), (103), (112). The average crystallite size (*d*) of the nanomaterials was calculated using Scherrer’s formula of peak width:*d = (0.9 λ)/β cosθ*(1)
where *λ* is the wavelength of X-ray used (0.15418 nm in the present case), *β* is the full width under radiation at half-maximum of the peak, and *θ* is the Bragg angle of the X-ray diffraction peak. The calculation gave a value of 5 ± 3 nm for the average crystallite diameter of the all as-prepared ZnS and ZnS:Mn nanowires. The crystal lattices of ZnS NCs suggest that all the samples are semicrystalized and the type of the heating used in the synthesis does not influence the crystal structure of ZnS NCs. In addition, there is no characteristic peak relevant to another phase, which demonstrates the high purity of the product. The influence of the amount of manganese on the XRD patterns was shown in Figure 1a. On those patterns, a very small shift of the diffraction peaks of (100), (002), and (001) can be observed. All Mn-doped ZnS NCs stabilized with EN showed a very slight shift towards the higher 2θ side compared with undoped ZnS due to the presence of Mn^2+^ instead of Zn^2+.^ The position of (002) peak for ZnS_En_Mn0.02 (1.0622 of Mn wt.%, Table 1), for example, is at 28.84° whereas the position of (002) peak for un-doped ZnS_EN NCs is at 28.60°. It is known that the ion radius of Mn^2+^ cation is bigger than the ion radius of Zn^2+^ cation. This peak position shift may occur due to the changes that occur in the lattice parameters of the host lattice on the incorporation of the dopant atoms [26]. Those changes can be observed on the XRD patterns of the samples with the increase of the manganese content. The further increase of the exact amount of Mn in the range 0.0944 wt.% (for ZnS_EN_Mn0.001) to 1.0622 wt.% (for ZnS_En_Mn0.02) affects also the very small changes in the XRD patterns of the samples, which are comparable. This effect can be also observed in the case of ZnS:Mn NCs stabilized with hydrazine (please see Appendix A). For ZnS:Mn NCs stabilized with hydrazine this 2 θ shift upon Mn doping in nanocrystals is very well observed. The position of (002) peak for ZnS_NH, ZnS_K_Mn0.01 (0.3857 of Mn wt.%, Table 1) and ZnS_NH_Mn0.01 (0.6847 of Mn wt.%, Table 1) is at 28.16°, 28.52° and 28.66°, respectively. This shift position of the XRD patterns can demonstrate the successfully Mn doping in ZnS NCs which is more efficient in the case of the ZnS NCs stabilized with hydrazine than ZnS NCs stabilized with ethylenediamine. These results are in good agreement with the results of MP-AES analysis of nanomaterials showed the exact doping contents of Mn in ZnS nanocrystals (see Table 1). Figure 1b shows the impact of the type of the used heat of source on the XRD patterns of ZnS:Mn samples stabilized with hydrazine. The similar position of the XRD patterns of all samples can indicate that the type of heating technique does not influence significantly the crystal structure of the NCs. In Figure 1c the influence of the used stabilizer on the XRD patterns was shown. The synthesis of the ZnS NCs in the presence of both stabilizers (ethylenediamine and hydrazine) leads to the formation of the NCs with the typical wurtzite structure. However, it can be observed that the relative peak ratio of (002) to these of (100) and (001) decreased when NH was used as a stabilizer instead of EN. Additionally, very small differences in the peak position can be observed. For example, the position of (022) peak for ZnS_NH is 28.10° whereas the position of (002) peak for ZnS_EN NCs is at 28.60°. In the case of the sample synthesized in the presence of hydrazine (Figure 1c) the peaks (101), (102), and (112) visible on the recorded pattern are higher in comparison to those present on the XRD of the sample prepared with the addition of ethylenediamine. This can be attributed to affected of the presence of bigger crystals in the case of ZnS NCs stabilized with NH. This result is in good agreement with the results of TEM analysis of nanomaterials (see Figure 2).

The morphology of the synthesized NCs was studied by transmission electron microscopy (TEM). Figure 2 shows the TEM images of Mn-doped ZnS NCs with the same Mn nominal concentration (the molar ratio of used Zn/S/Mn precursor was 1:2:0.01, see Table 1) stabilized with ethylenediamine (Figure 2a, sample ZnS_K_EN_Mn0.01) and hydrazine (Figure 2b; sample ZnS_K_NH_Mn0.01) prepared in conventional heating as well as in the microwave reactor (Figure 2c,e, sample ZnS_EN_Mn0.01; and Figure 2d,f, sample ZnS_NH_Mn0.01), respectively. We can observed that all nanocrystals are composed of many one-dimensional-like ZnS:Mn NCs. In the high magnification images (Figure 2e,f), it can be clearly seen that all ZnS:Mn nanowires with about 15 ± 20 nm diameter and up to 100 ± 170 nm length are composed of a large number of small crystallites. Additionally these ZnS nanowires have a tendency to agglomerate into bigger 3D flower-like hierarchical structures (see Figure 2a,c,d). Additionally, it seemed that ZnS:Mn stabilized with ethylenediamine and prepared in MW-assisted reaction (sample ZnS_EN_Mn0.01) exhibited a similar structures with a sample prepared using conventional heating (sample ZnS_K_EN_Mn0.01). However, the product is an agglomeration of one-dimensional nanowires with a longer length. In turn, the morphology of the samples ZnS_K_NH_Mn0.01 is different when compared to the sample ZnS_NH_Mn0.01 synthesized in MW. In the case of the ZnS_K_NH_Mn0.01 nanocrystals are larger and less regular in comparison with the ZnS_NH_Mn0.01 NCs synthesized in MW-assisted syntheses. Moreover, the crystals of ZnS:Mn NCs stabilized with hydrazine are less regular and bigger than the NCs stabilized with EN in the same conditions and are a mixture of 1D and 2D NCs.

Figure 3 presents the corresponding EDS mapping of the sample ZnS_EN_Mn0.01. Figure 3 confirms that ZnS:Mn NWs contain zinc, sulfur and Mn-doped elements evenly distributed in the crystal lattice of ZnS. The Mn concentration (wt.%) all ZnS:Mn NCs was determined with the microwave plasma atomic emission spectrometry (MP-AES). The results are presented in Table 1. In the case of the Mn-doped ZnSe NCs samples, the successful doping process of the manganese element in the crystal lattice of ZnS was observed with the loading percentage of about 0.0945, 0.1682, 0.4705, 1.0622, 0.3706, 0.6847, 0,1207 and 0.3857 wt. % for ZnS_EN_Mn0.001, ZnS_EN_Mn0.005, ZnS_EN_Mn0.01, ZnS_EN_Mn0.02, ZnS_NH_Mn0.001, ZnS_NH_Mn0.01, ZnS_K_EN_Mn0.01 and ZnS_K_NH_Mn0.01, respectively. In Table 1, it can be seen that regardless of the stabilizer and heating methods used, if the amount of Mn^2+^ precursor in the final reaction solution is increased, the concentration of the Mn^2+^ incorporated in the nanocrystals increases as well. However, the difference between the nominal and final doping contents in the NCs was observed, which is typical for solution-based synthesis approaches [27,28]. Sample ZnS_EN_Mn0.02 has a relatively high Mn^2+^ concentration (1.0622 wt.%) incorporated in the ZnS nanocrystals compared to sample ZnS_EN_Mn0.001 (0.0945 wt.%). The observed differences indicate that doping during the synthesis of semiconductor NCs using the solvothermal process is very sensitive to very small deviations in the reaction conditions.

For semiconductor NCs which are typically grown using colloidal synthesis, the doping concentrations attained in the experiments are much lower than expected from this limit. The likely reason is that the thermal equilibrium, which requires facile diffusion, may be far from being realized.

All colloidal reactions are challenging when it comes to maintaining high control of both the stoichiometry of the solid solution and the doping content in NCs [29]. Moreover, these results also indicated that the natures of stabilizers can result in different effectiveness of the Mn doping in ZnS. NH seemed to be more effective for Mn doping in ZnS compared to EN; 0.4705 Mn wt.% of ZnS_EN_Mn0.01 vs 0.6847 Mn wt.% of ZnS_NH_Mn0.01. In addition, MW was more effective than conventional heating regardless of the types of stabilizers.

One of the most important features influencing the photocatalytic activity of the tested samples is the development of their specific surface area. Therefore, detailed studies were carried out to determine the effect of the synthesis conditions on the size of the specific surface area. The N_2_ adsorption–desorption isotherms (Figure 4a–c) of ZnS and ZnS:Mn stabilized by EN and N_2_H_2_ samples exhibit a typical type-IV isotherm with a distinct hysteretic loop. According to the de Boer classification all of the hysteresis loops can be classified as a D type. This kind of hysteresis loop can be observed for the adsorbents with the irregular slit pores.

The comparison of samples with a similar, actual manganese concentration of about 0.38 wt.%, assigned as ZnS_NH_Mn0.001 (0.3706 wt.% Mn), ZnS_K_NH_Mn0.01 (0.3857 wt.% Mn) the relevant values of specific surface area and average pore width have been compared in Table 1 allows us to conclude that the key factor influencing the size of the surface is the selection of heating method. For conventionally heated samples, the specific surface area is below 190 m^2^/g, regardless of the used template. In addition, conventionally heated samples possess smaller intercrystalline spaces, which indicates a tighter packing of the structure and may affect the rate of dye diffusion to the catalyst surface. Another important factor influencing the development of the specific surface area is the type of used stabilizer. Samples for the synthesis of which ethylenediamine was used have a higher specific surface in comparison to those obtained with the application of hydrazine with the simultaneous use of microwave radiation as a heat source. For example, the specific surface area of ZnS_EN is 218.4 m^2^/g (with pore size of 8.76 nm) whereas for ZnS_NH is only 207.5 m^2^/g (with pore size of 5.63 nm). The opposite trend can be observed for samples that were obtained conventionally. At the same time, samples obtained with a longer template are less compact. What can be attributed to the larger size of intracrystalline spaces, which were measured on the basis of the mean pore size, which in some cases exceeds 9 nm for samples obtained with the use of ethylenediamine.

Further evidence for the properties of the ZnS:Mn samples were obtained with FT-IR spectroscopy. Figure 4d presents the FT-IR spectra of ZnS_EN_Mn0.01 stabilized with ethylenediamine and ZnS_NH_Mn0.01 stabilized with hydrazine. The FT-IR spectra look similar, which indicate the presence of two amines EN and N_2_H_4_ containing similar amino -NH_2_ groups on the surface of the NCs, respectively. The characteristic stretching vibrations for both samples at 1638 cm^−1^ and 3443 cm^−1^ are attributed to peaks for scissoring and stretching vibrations of the amine group. The peak at ν = 1142 cm^−1^ corresponds to the stretching of C-N bond and the two characteristic stretching vibrations at 670 cm^−1^ are from the C-H bond [30,31]. It is clear that the absorption band position at ν = 2841 cm^−1^ and 2924 cm^−1^ correspond to asymmetrical stretching vibrations of the methylene group is only visible for ZnS_EN_Mn0.01 sample. These results indicate the existence of both bidentate ligands like ethylenediamine and hydrazine on the NC surface, respectively. It was shown that these molecules had a significant impact on the anisotropic growth behavior of the NCs. Due to their polar character, they increase the solubility of the reagents during the synthesis. Additionally, due to their complexing properties, they can play a significant role in the reaction mechanism [31,32].

The optical properties of the prepared ZnS and ZnS:Mn nanowires were determined with PL and UV-Vis diffuse reflectance spectra. Figure 5a,b present DR UV-Vis spectra for samples with the different amounts of the Mn^2+^ ions synthesized MW heating and in the presence of ethylenediamine as a stabilizer; b) samples stabilized in the presence of the ethylenediamine and the hydrazine with the assistance of the MW heating; c) samples synthesized in the presence of the hydrazine with the similar actual Mn doping levels (about 0.38 wt.% Mn^2+^) with the assistance of traditional heat source (K) and the MW heating. One of the important features influencing the photocatalytic activity of the tested samples is the light absorption abilities of nanocrystals in the UV spectrum near 254 nm since the photocatalytic activities of various samples were examined with a UV lamp with irradiation at 254 nm. All samples were characterized with a strong and broad absorption band in the UV spectrum of light with the maximum at about 300 nm for ZnS and ZnS:Mn stabilized with hydrazine and at 320–330 nm for ZnS and ZnS:Mn stabilized with EN. The comparison of samples with a similar, actual manganese concentration of about 0.38 wt.%, assigned as ZnS_NH_Mn0.001 (0.3706 wt.% Mn), ZnS_K_NH_Mn0.01 (0.3857 wt. % Mn) allows to conclude that one of the key factors influencing the UV absorbance is the selection of heating method. Sample ZnS_NH_Mn0.001 synthesized under MW-assisted reaction shows higher the light absorption abilities in the UV range of spectrum compare to sample synthesized with the assistance of traditional heat source. Another important factor influencing the optical properties of ZnS NCs is the type of used stabilizer. Figure 5b indicates that the un-doped ZnS nanocrystals stabilized with hydrazine absorb more efficient light in the UV light region of spectrum, including the irradiation at 254 nm. Moreover, the UV spectra also clearly indicate that the significant red shift in the UV spectrum into the visible spectrum of light can be observed for the NCs doped with Mn^2+^ ions in the crystal lattice. This phenomenon can be observed for both ZnS NCs stabilized with EN and N_2_H_2._ However, this effect can be more visible for ZnS:Mn, as it has EN ligands. Additionally, it can be observed that with the increased Mn concentration in the ZnS NCs, the absorption in the visible light increases due to the Mn dopants (Figure 5a). This enhanced absorbance to the visible region can be probably linked to the increase in defect sites in the crystal structure of the doped ZnS NCs [28,30,31]. In the UV light spectrum the situation is more complicated. In the case of ZnS_EN, the light absorption abilities first increased as the Mn doping level increased for samples ZnS_EN_Mn0.001 and ZnS_EN_Mn0.005 and again decreased as the Mn doping level kept increasing for samples ZnS_EN_Mn0.01 and ZnS_EN_Mn0.02.

UV-Vis diffuse reflectance spectra were used to determine the optical band gap for all ZnS and ZnS NCs, using the standard Kubelka-Munk function according to literature [33]. Spectrophotometric method is a rapid and precise technique for optical band gap determination for powder samples with the standard error about ±0.02 [33]. Figure 5c shows the plot of determination of the band gap for the ZnS_HN_Mn0.01 sample. Table 1 presents the calculated band gaps for all samples. The values are, accordingly, 3.30 eV for un-doped ZnS_EN, 2.93 eV for ZnS_EN_Mn0.001, 3.31 eV for ZnS_EN_Mn0.005, 2.94 eV for ZnS_EN_Mn0.01 and 2.72 eV for ZnS_EN_Mn0.02 with the highest content of Mn. As it can be seen, the band gap values of all ZnS and ZnS:Mn stabilized with ethylenediamine decrease when compared with those of ZnS bulk value materials (3.7 eV) [3,10,34]. However, the changes of band gap values upon the Mn doping levels were complicated, initially, the band gap decreased with an increase of Mn doping levels than increased for ZnS_EN_Mn0.005 and again decreased as the Mn doping level kept increasing for ZnS_EN_Mn0.01 and ZnS_EN_Mn0.02. In the case of ZnS and ZnS:Mn stabilized with hydrazine, we also observe slightly increased band gap values in comparison with the ZnS bulk value materials, i.e. 3.78 eV for ZnS_NH, 3.81 eV for ZnS_NH_Mn0.001. For ZnS:Mn with a higher concentration of Mn^2+^ ions in the NCs, we observe a slightly decreased band gap when compared to ZnS bulk value materials. Moreover, the optical band gap of samples ZnS_NH_Mn0.001 and ZnS_K_NH_Mn0.01 synthesized in the presence of hydrazine with the similar actual Mn doping levels (about 0.38 wt.% Mn^2+^, Table 1) with the assistance of the MW heating and traditional heat source, are 3.81 eV and 3.72 eV, respectively. This can suggest that the doping process of Mn atoms has not so high impact on the engineered ZnS band gap like type of used ligands or type of used heating methods. Its looks that the type of used stabilizer has more significant impact on the changes of optical band gap of nanocrystals. For example, the calculated optical band gap for ZnS_NH is 3.78 eV whereas for ZnS_EN samples was 3.30 eV. And the selection of different stabilizers did not induce a significant size difference in synthesized ZnS samples if MW was used instead of conventional heating as implied by TEM analysis results (Figure 2).

The room-temperature photoluminescence (PL) spectra of the as-prepared ZnS and Zn:Mn NCs is presented in Figure 6. All emission spectra were measured for an excitation wavelength at 310 nm. Figure 6 presents PL emission spectra for ZnS and ZnS:Mn NCs of ZnS and ZnS:Mn NCs (a) samples with the different amount of the Mn^2+^ ions synthesized MW heating and in the presence of ethylenediamine as a stabilizer; (b) samples stabilized in the presence of the ethylenediamine and the hydrazine with the assistance of the MW heating and (c) samples synthesized in the presence of the hydrazine including the samples with the similar actual Mn doping levels (about 0.38 wt.% Mn^2+^) with the assistance of traditional heat source (K) and the MW heating. The PL spectra of all the Mn doped ZnS samples stabilized with EN or N_2_H_2_ were characterized with one sharp band at about 590 nm and a broad peak with the maximum between at 420–470 nm. The band at about 590 nm is attributed to ^4^T_1_-^6^A_1_ transition in the Mn^2+^ ions [28,30]. The higher concentration of the Mn^2+^ ions in the Zn:Mn samples led to the increased intensity of emission band corresponding to transition in the Mn^2+^ ions (Figure 6a,b).

The comparison of samples with a similar, actual manganese concentration of about 0.38 wt.%, assigned as ZnS_NH_Mn0.001 (0.3706 wt.% Mn), ZnS_K_NH_Mn0.01 (0.3857 wt.% Mn) allows concluding that the selection of heating method has also impact on PL emission of ZnS:Mn NCs. Samples prepared in MW reactor show higher PL emission of the band at 590 nm. On the other hand, Figure 6b shows that the type of used stabilizer for the synthesis of un-doped ZnS NCs does not impact the PL properties of nanocrystals. As can be seen, the emission of both samples is very low, compare to Mn-doped ZnS NCs. As we know, the PL emission spectra can be also useful to explain the variation of photocatalytic activities among the samples since they can provide information on the electron-hole separation efficiency. In most cases, the higher the PL intensity from the sample indicated a higher probability of electron-hole recombination, i.e., lower electron-hole separation efficiency which mostly decreases the photocatalytic activities. However, in the cease of Mn-doped ZnS NCs, the emission which can be observed is coming mostly from electron-hole recombination in Mn ions (_4_T^1^-_6_A^1^) not only from the sample surface but also from nanocrystals (Figure 3c). In addition, the exact Mn doping concentration in ZnS NCs is still relatively low.

### 3.2. Test of Photocatalytic Activity

The photocatalytic activities of the as-synthesized ZnS and ZnS:Mn nanowires stabilized with ethylenediamine and hydrazine were evaluated by the photodegradation of rhodamine B under UV irradiation. It is known that the photocatalytic process takes place on the surface of catalysts. Therefore, the adsorption–desorption equilibrium measurements were performed before irradiation to investigate the adsorption ability of the samples. First, the suspensions were magnetically stirred for 1 hour in dark to establish the adsorption–desorption equilibrium between the RhB and the photocatalyst. In the dark experiment the solution phase concentration of RhB rapidly decreases upon addition to the ZnS or ZnS:Mn NCs suspension due to adsorption. About 15% of RhB was adsorbed in the dark due to the a relatively high BET surface of used photocatalysts. The characteristic absorption of RhB at 550 nm was selected to monitor the photocatalytic degradation process. Figure 7a presents the dependence of the RhB-normalized UV absorption at the maximum of absorbance versus illumination time in the presence of the ZnS_NH_Mn0.01 sample. It can be seen that over 60% of the dye was degraded after 3 h while the RhB was not degraded at all in the solution without the presence of ZnS NCs photocatalysts (Figure 7b, the blank test). Figure 7c,d present the time profiles of the photocatalytic efficiency of the degradation process of RhB in a water solution versus irradiation time under UV light in the presence of ZnS and ZnS:Mn NCs stabilized with ethylenediamine and ZnS and ZnS:Mn stabilized with N_2_H_2_, respectively. All ZnS and ZnS:Mn NCs show the photocatalytic activity in the degradation of RhB. In most cases, the efficiency of the photocatalytic degradation process of RhB increases with the increase of irradiation time.

The % of dye degradation is calculated using the following equation:% Degradation= (A_o_ − A_t/_A_o_) × 100(2)
where A_o_ is the absorbance at time t = 0 min and A_t_ is the absorbance after time t min of treatment, Ao and A_t_ are recorded at λ_max_ of dye (at 550 nm for RhB) [35]. Figure 8 presents the photocatalytic degradation kinetics of photocatalytic degradation process of RhB in aqueous solution under UV light with the presence of ZnS and ZnS:Mn NCs stabilized with EN and N_2_H_2_, respectively.

#### 3.2.1. Effect of Mn-Doping on the Photocatalytic Activity 

In order to check the doping effect on the photocatalytic performance in the RhB degradation process, the ZnS NCs stabilized with ethylenediamine were prepared with varied Mn doping concentration (see Table 1). Photocatalytic studies have been carried out for ZnS NCs and different Mn percentages of ZnS:Mn NCs stabilized with EN for the photodegradation of RhB under UV light irradiation. The time profiles and photocatalytic degradation kinetics as well as the comparison of the photocatalytic activity of ZnS NCs in the RhB degradation process for different Mn concentration are shown in Figure 7a, Figure 8a and Figure 9a.

The study shows that the addition of Mn strongly improves the photocatalytic properties. As shown in Figure 7a and Figure 9a, the photocatalytic activity of RhB is about 45%, 36%, 27%, 18%, 16% for ZnS_En_Mn0.02 (1.0622 of Mn wt.%), ZnS_EN_Mn0.1 (0.4705 of Mn wt.%), ZnS_EN_Mn0.005 (0.1682 of Mn wt.%), ZnS_EN_Mn0.001 (0.0944 of Mn wt.%), ZnS_EN, respectively, under UV light irradiation within 300 min. The ZnS_EN_Mn0.02 sample displays the highest photocatalytic activity and about 45% of RhB dye molecules can be degraded within 300 min. Additionally, for all the samples of the ZnS and ZnS:Mn NCs, a pseudo-first-order model was applied and the corresponding equation ln(C_0_/C) = kt was used, where C_0_ and C are the concentrations of dye in the solution at times corresponding to 0 and t, respectively, and k is the pseudo-first-order rate constant. The correlation with regard to pseudo-first-order reaction kinetics was 0.9913, 0.9434, 0.9681, 0.9303, and 0.9496 for the ZnS_EN_Mn0.02, ZnS_EN_Mn0.01, ZnS_EN_Mn0.005, ZnS_EN_Mn0.001 and ZnS_EN samples, respectively, as shown in Figure 8a.

The comparisons of the reported photodegradation efficiency of ZnS:Mn stabilized by EN photocatalyst synthesized with the microwave method for the degradation of RhB are summarized in Figure 9a. Obviously, the ZnS_EN_Mn0.02 exhibited much smaller degradation efficiency compared to standard powder P25 TiO_2_ (see also Appendix A). Taking into account the positive impact of Mn doping on the activity in the photocatalytic degradation of RhB we need to consider also the exact Mn doping concentration and characterization in ZnS NCs. As can be seen in Table 1 the nominal Mn concentration is different than the exact doping amount of Mn. For instance, ZnS_EN_Mn0.01 (0.4705 of Mn wt.%) has different doping amount from ZnS_NH_Mn0.01 (0.6847 of Mn wt.%), ZnS_K_EN_Mn0.01 (0.1207 of Mn at%), and ZnS_K_NH_Mn0.01 (0.3857 of Mn at%). The photocatalytic activity of these samples in the degradation process of RhB was also different about 36%, 60%, 16% and 30% for ZnS_EN_Mn0.1, ZnS_NH_Mn0.01, ZnS_K_EN_Mn0.01 and ZnS_K_NH_Mn0.01, respectively. The highest photocatalytic activity presents sample ZnS_NH_Mn0.01 with the highest effective Mn doping concentration. One of the important parameters influencing the photocatalytic activity of the ZnS:Mn samples is the UV light absorption abilities, band gap properties as well as the specific surface area. All these properties have been discussed in detail in the previous chapters. Table 1 and Figure 5a show that the Mn doping led to a shift of the ZnS optical band gap from 3.30 to 2.72 eV. The band gap of ZnS_EN_Mn0.01, ZnS_NH_Mn0.01, ZnS_K_EN_Mn0.01 and ZnS_K_NH_Mn0.01 samples was 2.94 eV, 3.64 eV, 3.58 eV and 3.72 eV, respectively. On the other hand, the specific surface area of these samples was adequate of 207.4 m^2^/g, 211.2 m^2^/g, 185.1 m^2^/g and 194.8 m^2^/g for ZnS_EN_Mn0.01, ZnS_NH_Mn0.01, ZnS_K_EN_Mn0.01 and ZnS_K_NH_Mn0.01, respectively. Nanomaterials synthesized in MW-assisted methods show higher specific surface area and also higher light absorption abilities (Figure 5c). The Mn^2+^ ions doping caused also the enhanced absorbance to the visible region of the spectrum and PL emission at 590 nm. However, the Mn doping in ZnS NCs allows them to be used as effectual UV photocatalysts for the degradation of organic contaminants. This enhancement can suggest a more efficient separation process of the photogenerated charge carriers than that of un-doped ZnS, which can positively influence the photocatalytic activity even under UV light. The Mn^2+^ ions in ZnS NCs behave like electron sinks, which can effectively trap and transfer the photogenerated electrons, and then improve the separation of electrons and hole pairs. Similar effect has been observed also for other nanomaterials, e.g. ZnSe, ZnO, CuO, by other authors [30,36,37].

#### 3.2.2. Effect of the Stabilizing Agent and the Used Synthetic Method on the Photocatalytic Activity

In order to check the effect of the used stabilizing agent and the synthetic process on the photocatalytic performance of ZnS NCs in the RhB degradation process, the ZnS NCs stabilized with ethylenediamine and hydrazine were prepared with the same Mn doping concentration using both the microwave and the conventional heating (see Table 1). Likewise, the photocatalytic studies have been carried out for all ZnS-based samples for the photodegradation of RhB under UV light irradiation. The time profiles and photocatalytic degradation kinetics as well as the comparison of the photocatalytic activity of ZnS_EN, ZnS_EN_Mn0.01, ZnS_NH and ZnS_NH_Mn0.01, ZnS_K_NH_Mn 0.01 in the RhB degradation process are presented in Figure 7a,b, Figure 8a,b and Figure 9b,c. The study shows that the use of the hydrazine as a stabilizing agent strongly improves the photocatalytic properties in both cases of Mn-doped ZnS and un-doped ZnS NCs. The highest efficiency of the degradation process was observed for sample ZnS_NH_Mn0.01 (0.6847 of Mn wt.%), where during 3 h of UV irradiation, 61% of the dye was degraded. When we compare the un-doped ZnS NCs synthesized in MW reactor, samples ZnS_NH and ZnS_EN (Figure 9b) we observe higher photocatalytic performance for the crystals with shorter amine molecules as hydrazine on the surface of the NCs. The specific surface area of ZnS_EN was 218.4 m^2^/g (with pore size of 8.76 nm) whereas for ZnS_NH was only 207.5 m^2^/g (with a pore size of 5.63 nm). However, Figure 5b shows that the un-doped ZnS nanocrystals stabilized with hydrazine (ZnS_NH) have better light absorption abilities in the UV light region of the spectrum than ZnS NCs stabilized with ethylenediamine, including the irradiation at 254 nm. These properties make ZnS_NH NCs a more efficient photocatalyst (almost twice, 16% for ZnS_EN vs. 32% for ZnS_NH). 

It is well known that microwave heating is a promising technology and its applications have been fast growing due to its exclusive effects, such as rapid volumetric heating, increased reaction rates, shortened reaction time and improved reaction selectivity [25]. In our study, we observed much better photocatalytic performance of the NCs synthesized in MW-assisted synthesis when compared to the sample synthesized in the reactor with conventional heating. For example, only 30% of the RhB molecules were photocatalytically reduced with sample ZnS_K_NH_Mn0.01 (the surface area = 194.8 m^2^/g), having the same nominal Mn doping concentration and stabilizing agent like sample ZnS_NH_Mn0.01 (the surface area = 207.4 m^2^/g), where 61% of the dye was degraded. In our study, microwave irradiation was effectively used to modify the desired morphology and high surface areas of ZnS-based NCs (see Table 1 and Figure 2). These properties have huge effects on the photocatalytic properties of nanocrystals [30]. In turn, when we compare the ZnS:Mn NCs with the similar actual Mn doping levels about 0.38 wt.% Mn (Table 1, Figure 9c) in NCs (ZnS_NH_Mn0.001 versus ZnS_K_NH_Mn0.01) synthesized in the presence of hydrazine with the assistance of the MW heating and traditional heat source, we observe higher photocatalytic performance for the crystals synthesized in microwave-assisted synthesis. Almost 48% of RhB molecules were photocatalytically reduced with sample ZnS_NH_Mn0.001 having the similar Mn doping concentration and hydrazine as the stabilizing agent. In turn, only about 30% of RhB molecules were photocatalytically reduced in the presence of ZnS_K_NH_Mn0.01 samples. The specific surface area of ZnS_K_NH_Mn0.01 was 194.8 m^2^/g (with pore size of 7.73 nm) whereas for ZnS_NH_Mn0.001 was 207.4 m^2^/g (with a pore size of 8.76 nm). 

In fact, the ZnS_NH_Mn0.01 sample shows the highest efficiency in the degradation process of all catalysts but its pseudo-first-order rate constant was clearly lower than those of some other catalysts. The pseudo-first-order rate constant was 0.9496, 0.9434, 0.9402, 0.7842 and 0.9577 for the ZnS_En, ZnS_EN_Mn0.01, ZnS_NH and ZnS_NH_Mn0.01, ZnS_K_NH_Mn 0.01 samples, respectively, as shown in Figure 8. It can be found that the experimental data were not ideally fitted to the pseudo-first-order model, with low correlation coefficients of 0.7842 for ZnS_NH_Mn0.01. This indicates that the experimental data did not well obey the pseudo-first-order kinetic model. It is known that photocatalytic oxidation rate constant, k, is a property of a photocatalyst, and does not significantly depend on reactant structure [38]. Therefore, this deviation can be connected to different aspects related to the surface properties of photocatalysis as well as the reaction mechanism including the adsorption/diffusion process; e.g. the photocatalyst surface was saturated in dye over nearly the entire concentration range studied. However, in order to give a clear explanation of this deviation, further research of the system, including an understanding of the adsorption mechanisms and the detailed kinetic analysis, is required [39,40]. 

Figure 10 shows the incident photon-to-electron conversion efficiencies (IPCE) as a function of wavelength and potential of the ZnS_NH_Mn0.01 sample. This presents the best efficiency photoactivity. The IPCE values for different wavelengths were derived with determination of power of the incident monochromatic beam at each of the wavelength, using the following equation:IPCE(%) = (1240/ λ) × (l_ph_/p_in_) X 100(3)
where λ is the wavelength of the incident light in nm, l_ph_ is the photocurrent in A cm^−2^ and p_in_ is the power of the incident monochromatic light in Wcm^−2^ [30,40]. The ZnS_NH_Mn0.01 shows the photocurrent response mostly in the UV range of the light spectrum, with the optimum IPCE efficiency near 5% at about 380 nm and 1450 mV. In turn, the IPCE of P25 TiO_2_ (see also Appendix A) is 57% according to literature [41].

It is well known that the practical application of the photocatalyst strongly depends also on the stability and recyclability of the photocatalysts. Therefore, to study the performance stability of the sample which showed the best photocatalytic activity, the recycling test of photocatalytic degradation of RhB was performed under UV light. Figure 11 shows the stability tests for the photodegradation of RhB over the ZnS_NH_Mn0.01, photocatalysts under UV light. The stability test of the photocatalysts was performed by cycling experiments. It can be clearly seen that the efficiency of the photocatalytic process is relatively stable, however the efficiency of photodegradation decreased only slightly. These results indicate that ZnS is a very good candidate for a photocatalyst in the degradation of water pollutants.

## 4. Conclusions

In summary, ZnS and ZnS:Mn nanowires were successfully obtained in a microwave-assisted solvothermal synthesis method, which is simple, cost effective and easily scalable. Two molecules were used as ligands: ethylenediamine and hydrazine. These ligands were used as capping agents to tune the morphology of the ZnS samples by influencing the crystal growth. The morphology and optical properties of the as-prepared NCs were characterized with the XRD, TEM, DR UV-Vis, FT-IR and BET analyses. The as-synthesized ZnS:Mn NCs stabilized by hydrazine showed the largest photocatalytic activity in the degradation of rhodamine B. The enhanced photocatalytic performance could be attributed to a large specific surface area, high Mn^2+^ doping concentration and short amines on the surface as ligands, which improved the migration rate of the electron/hole to the surface NCs. It has been proven that the photocatalytic activity of the Mn-doped ZnS NCs strongly depends on the dopant concentration. The as-designed ZnS and ZnS:Mn nanowires are promising in degrading organic pollutants in water.

## Figures and Tables

**Figure 1 materials-14-05840-f001:**
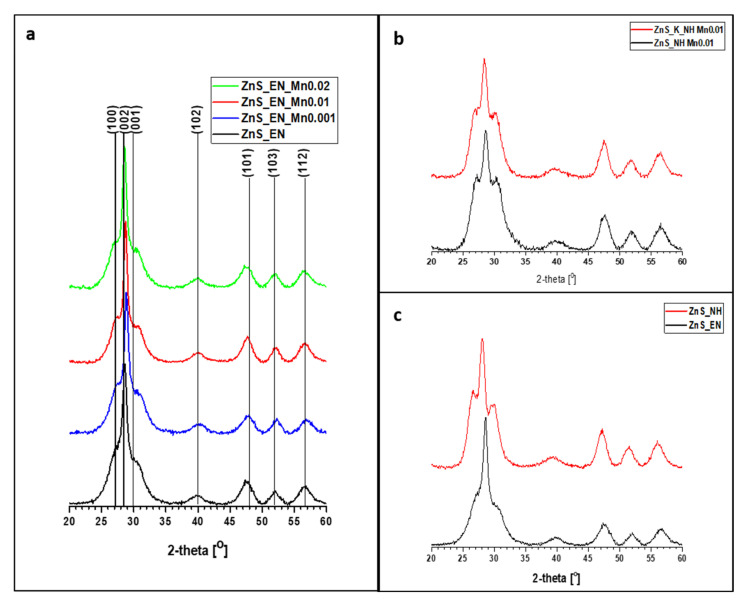
XRD patterns of ZnS samples prepared (**a**) with a different content of manganese and ethylenediamine as a stabilizer (**b**) with different types of using the heat source (**c**) with a different stabilizer; the peaks in the pattern matching well with wurtzite structure according to JCPDS#96-110-0045.

**Figure 2 materials-14-05840-f002:**
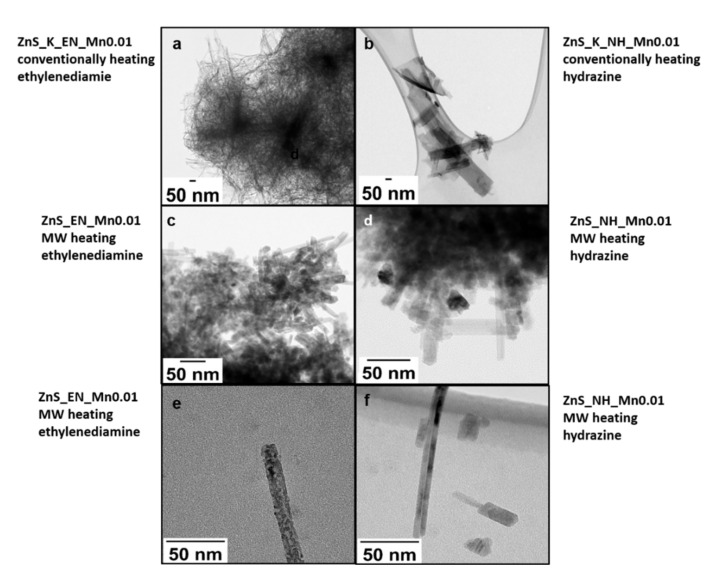
TEM images of the ZnS_Mn 0.01 (**a**) conventionally heated in the presence of ethylenediamine; (**b**) conventionally heated in the presence of hydrazine; (**c**,**e**) microwave heated in the presence of ethylenediamine as a stabilizer (different magnifications; (**d**,**f**) microwave heated in the presence of hydrazine as a stabilizer (different magnifications).

**Figure 3 materials-14-05840-f003:**
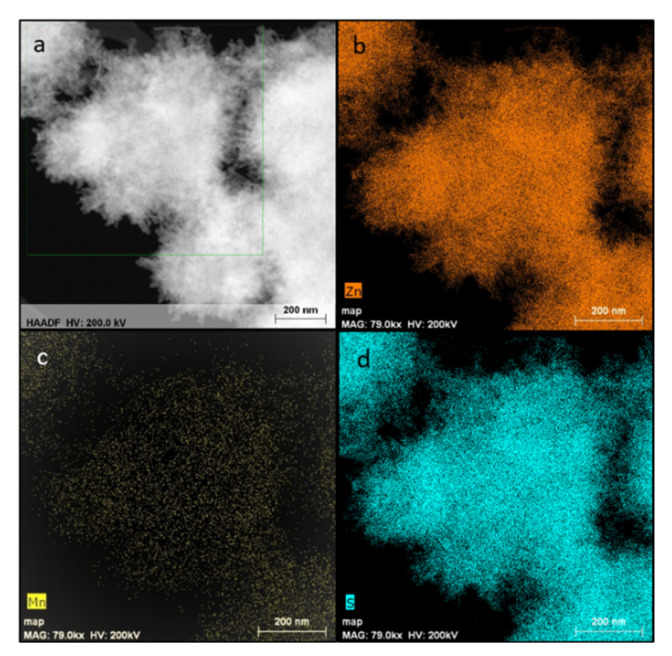
HAADF-STEM (**a**) imaging study and EDX analysis of chemical composition of ZnS_EN_Mn0.01 including zinc (**b**), manganese elements (**c**), sulphur (**d**).

**Figure 4 materials-14-05840-f004:**
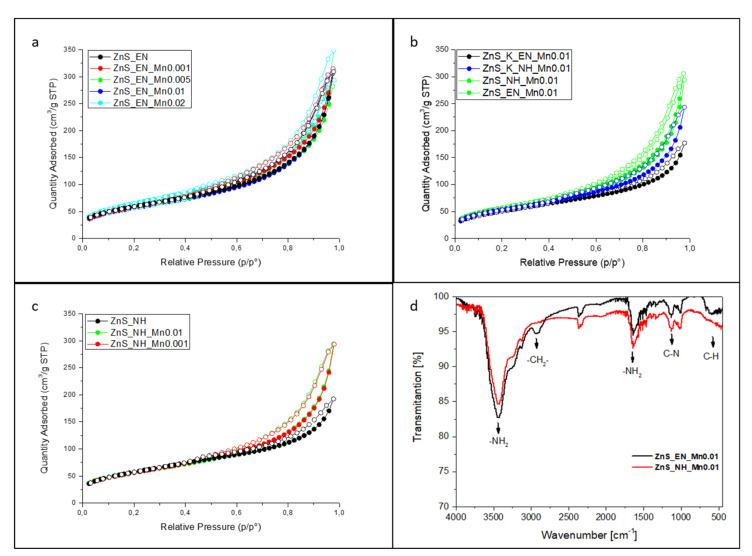
N_2_ adsorption–desorption isotherms for ZnS and ZnS:Mn NCs; (**a**) samples with the different amount of the Mn^2+^ ions synthesized MW heating and in the presence of ethylenediamine as a stabilizer; (**b**) comparison of the sorption isotherms of the samples synthesized in the presence of the ethylenediamine and the hydrazine with the assistance of traditional heat source (K) and the MW heating; (**c**) sorption isotherms of the samples with a different Mn^2+^ content synthesized in the presence of hydrazine as a stabilizer with a assistance of microwaves; (**d**) FT-IR spectra of ZnS_EN_Mn0.01 stabilized with ethylenediamine and ZnS_NH_Mn0.01 stabilized with hydrazine.

**Figure 5 materials-14-05840-f005:**
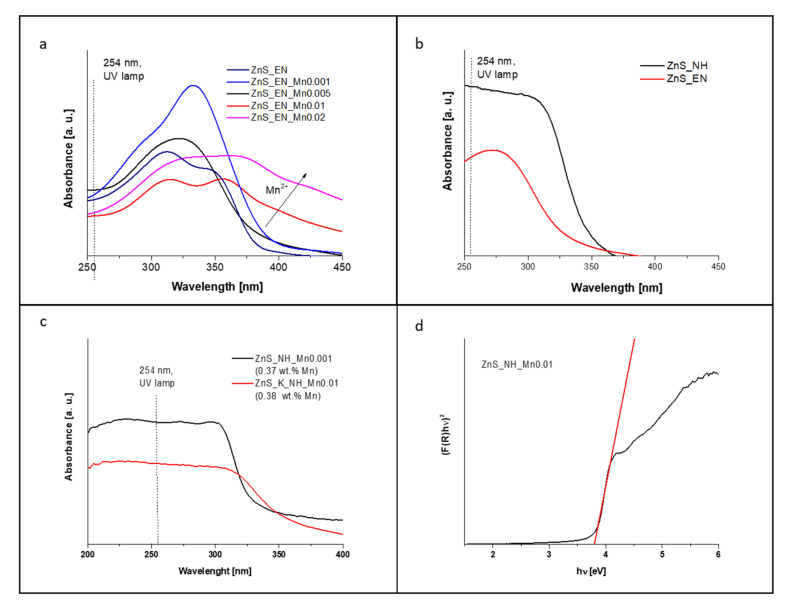
DR UV-Vis spectra of ZnS and ZnS:Mn NCs (**a**) samples with the different amount of the Mn^2+^ ions synthesized MW heating and in the presence of ethylenediamine as a stabilizer; (**b**) samples stabilized in the presence of the ethylenediamine and the hydrazine with the assistance of the MW heating; (**c**) samples synthesized in the presence of the hydrazine with the similar actual Mn doping levels (about 0.38 wt.% Mn^2+^ ) with the assistance of traditional heat source (K) and the MW heating; (**d**) the plot of determination of the band gap for the ZnS_HN_Mn0.01 sample.

**Figure 6 materials-14-05840-f006:**
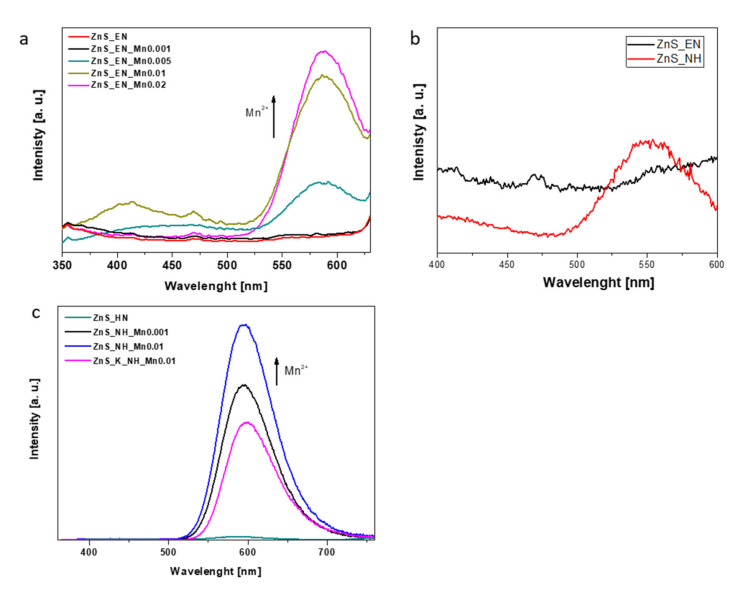
PL emission spectra for ZnS and ZnS:Mn NCs of ZnS and ZnS:Mn NCs (**a**) samples with the different amount of the Mn^2+^ ions synthesized MW heating and in the presence of ethylenediamine as a stabilizer; (**b**) samples stabilized in the presence of the ethylenediamine and the hydrazine with the assistance of the MW heating; (**c**) samples synthesized in the presence of the hydrazine including the samples with the similar actual Mn doping levels (about 0.38 wt.% Mn^2+^) with the assistance of traditional heat source (K) and the MW heating.

**Figure 7 materials-14-05840-f007:**
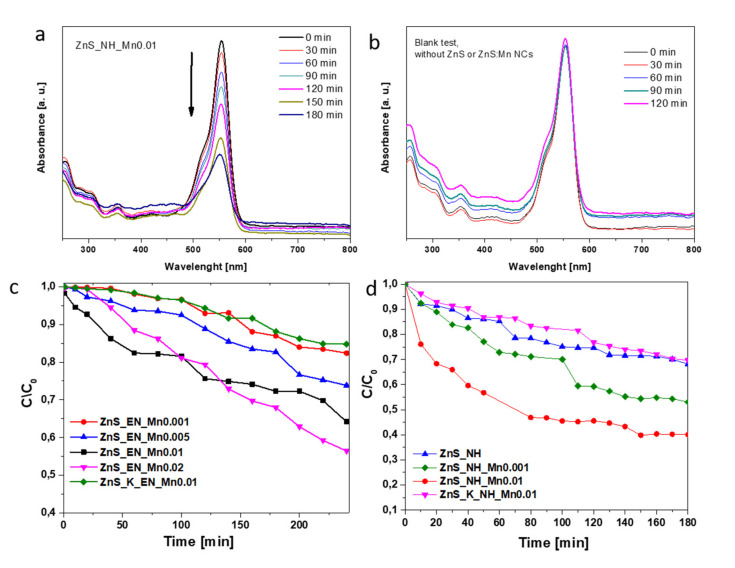
Time-dependent UV-Vis absorption spectra for RhB observed during incubation with the ZnS_NH_Mn0.01 samples under UV illumination, the concentration of dye RhB vs ZnS based NCs was 1:67 (typical 0.15 mg RhB vs 10 mg of ZnS:Mn NCs) (**a**), blank test (**b**), time profiles of photocatalytic degradation process of RhB in water solution under UV light with the presence of ZnS and ZnS:Mn NCs stabilized with EN (**c**) and ZnS and ZnS:Mn stabilized with N_2_H_2_ (**d**).

**Figure 8 materials-14-05840-f008:**
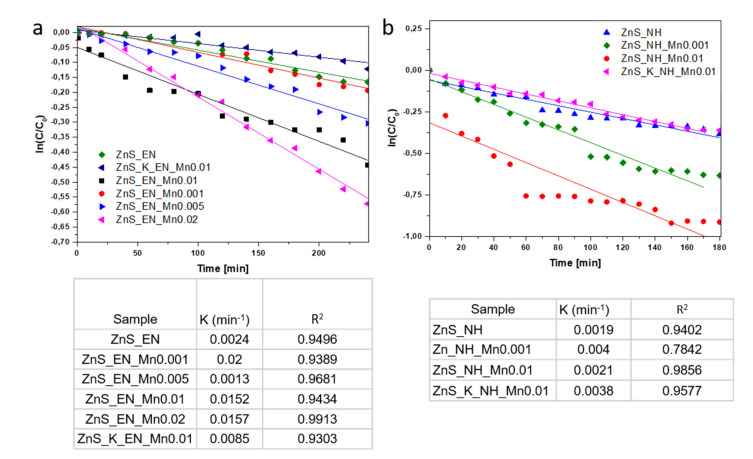
Photocatalytic degradation kinetics of photocatalytic degradation process of RhB in aqueous solution under UV light with the presence of ZnS and ZnS:Mn NCs stabilized with EN (**a**) and ZnS and ZnS:Mn stabilized with N_2_H_2_ (**b**).

**Figure 9 materials-14-05840-f009:**
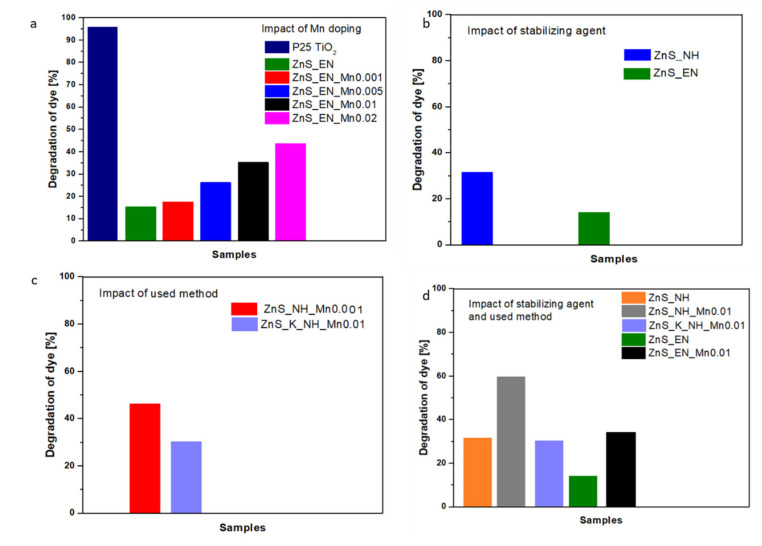
The comparison of efficiency of photocatalytic degradation of RhB for (**a**) different ZnS and ZnS:Mn NCs samples with the different amount of the Mn^2+^ ions synthesized MW heating and in the presence of ethylenediamine versus standard P25 TiO_2_ photocatalyst; (**b**) for ZnS samples stabilized in the presence of the ethylenediamine and the hydrazine with the assistance of the MW heating; (**c**) samples synthesized in the presence of the hydrazine including the samples with the similar actual Mn doping levels (about 0.38 wt.% Mn^2+^) with the assistance of traditional heat source (K) and the MW heating;. (**d**) different ZnS and ZnS:Mn NCs samples.

**Figure 10 materials-14-05840-f010:**
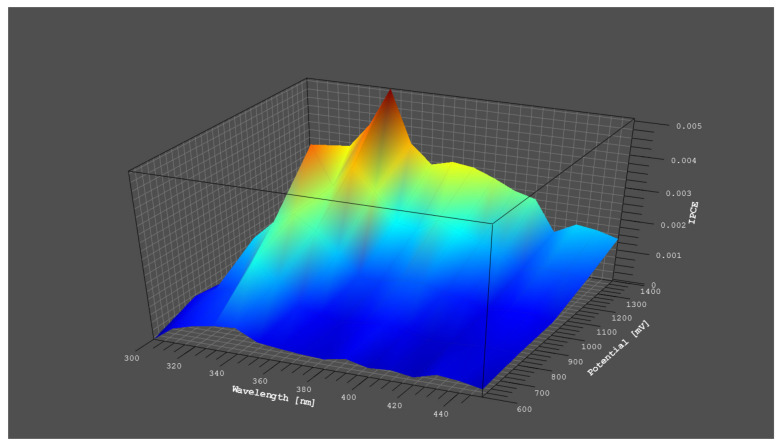
The incident photon-to-electron conversion efficiencies (IPCE) as a function of wavelength and potential for ZnS_NH_Mn0.01.

**Figure 11 materials-14-05840-f011:**
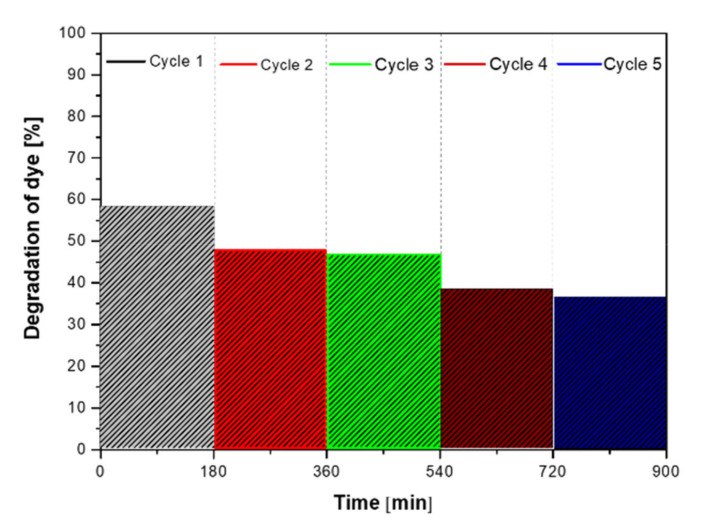
The stability test for the photodegradation process of RhB in the presence of ZnS_NH_Mn0.01 photocatalysts.

**Table 1 materials-14-05840-t001:** Detailed experimental parameters for synthesis of ZnS NCs.

Sample	Molar Ratio of Zn/S/Mn Precursor	Stabilizer	Bandgap [eV] (±0.02)	Surface Area [m^2^/g]	Average Pore Width [nm]	Mn Concentration [ppm]	Mn Concentration [wt.%]
**MW, 160–170 ^o^C, 8 bar, 20 min., stabilizer ethylenediamine (EN)**							
ZnS_EN	1:2:0	EN	3.30	218.4	8.76	–	–
ZnS_EN_Mn0.001	1:2:0.001	EN	2.93	216.24	9.00	944.97	0.0945
ZnS_EN_Mn0.005	1:2:0.005	EN	3.31	226.7	7.72	1682.43	0.1682
ZnS_EN_Mn0.01	1:2:0.01	EN	2.94	207.4	9.12	4705.26	0.4705
ZnS_EN_Mn0.02	1:2:0.02	EN	2.72	240.1	8.98	10622.04	1.0622
**MW, 160–170 ^o^C, 6–8 bar, 20 min., stabilizer hydrazine (N_2_H_4_)**							
ZnS_NH	1:1:0	N_2_H_4_	3.78	207.5	5.63	–	–
ZnS_NH_Mn0.001	1:2:0.001	N_2_H_4_	3.81	207.4	8.76	3706.19	0.3706
ZnS_NH_Mn0.01	1:2:0.01	N_2_H_4_	3.64	211.2	8.76	6847.25	0.6847
**Conventional heating, 160–170 ^o^C, 8 bar, 20 min.**							
ZnS_K_EN_Mn0.01	1:2:0.01	EN	3.58	185.1	5.92	1207.21	0.1207
ZnS_K_NH_Mn0.01	1:2:0.01	N_2_H_4_	3.72	194.8	7.73	3857.24	0.3857

## Data Availability

The data presented in this study are available on request from the corresponding author.

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
