# Peer review of "Photodegradation Process of Organic Dyes in the Presence of a Manganese-Doped Zinc Sulfide Nanowire Photocatalyst"

_materials, 2021, doi:10.3390/ma14195840_

Round 1

Reviewer 1 Report

This paper by Adam Zaba et al. deals with photocatalytic activities of various Mn-doped ZnS nanowires prepared with two different stabilizers towards decolorization of Rhodamine 6G solutions under UV light irradiation. The intention of this article may be of interest, but the overall quality is low to prevent me to recommend it to be published in Materials.

  1. As the authors mentioned, the main purposes of this article can be 1. Discuss the impact of stabilizing agents used during the synthetic process and 2. Impacts of Mn doping on the photocatalytic activities of synthesized ZnS nanowires. Those two topics seemed to be better discussed separately and more carefully with some of the selected samples. For instance, characterization and photocatalytic results of the sample series of ZnS_En_Mn (0 – 0.02) samples can be made to discuss the Mn doping effects on photocatalytic activity. Then results of ZnS_En_Mn0 and 0.01, ZnS_NH_Mn0 and 0.01, ZnS_K_EN or NH_Mn 0.01 samples can be compared to discuss the impacts of different synthetic processes (using MW and conventional heating with two stabilizing agents). Now the selection of sample series for comparison does not seem relevant to the main purposes of this article.
  2. The results of photocatalytic experiments show the variation of activities of samples depends on Mn doping levels and synthetic processes. However, the origin of those variations can be hardly correlated to other characterization results in the current version of the manuscript. For instance, DR UV-vis spectra show the increase of the visible light absorption upon the Mn doping, but it is confusing how the increase of visible light absorption can correlate the photocatalytic activity under the UV light irradiations.

Not only the sample series for comparison/discussion must be more carefully selected but also all the experimental datum must be much more carefully examined to deliver the main idea of this work by correlating the photocatalytic activities of various samples to analysis results.

Author Response

Response to Reviewer 1

We are very grateful to Reviewer for their comments and suggestions. The manuscript has been substantially reformatted and modified based on these suggestions. Detailed responses to each of the review’s comments are given below. Reviewer comments are in black and the responses to the reviewer comments are in blue, as well as the corrections in manuscript are in read.

Reviewer 1:

This paper by Adam Zaba et al. deals with photocatalytic activities of various Mn-doped ZnS nanowires prepared with two different stabilizers towards decolorization of Rhodamine 6G solutions under UV light irradiation. The intention of this article may be of interest, but the overall quality is low to prevent me to recommend it to be published in Materials.

1.As the authors mentioned, the main purposes of this article can be 1. Discuss the impact of stabilizing agents used during the synthetic process and 2. Impacts of Mn doping on the photocatalytic activities of synthesized ZnS nanowires. Those two topics seemed to be better discussed separately and more carefully with some of the selected samples. For instance, characterization and photocatalytic results of the sample series of ZnS_En_Mn (0 – 0.02) samples can be made to discuss the Mn doping effects on photocatalytic activity. Then results of ZnS_En_Mn0 and 0.01, ZnS_NH_Mn0 and 0.01, ZnS_K_EN or NH_Mn 0.01 samples can be compared to discuss the impacts of different synthetic processes (using MW and conventional heating with two stabilizing agents). Now the selection of sample series for comparison does not seem relevant to the main purposes of this article.

Thank you for this valuable remarks. According to Review’s suggestion the whole manuscript was revised, especially the experimental part, section 3.2 Test of photocatalytic activity.  The data was again selected and discussed in separated section. Please see the paragraphs 3.2.1 and 3.2.2.  We hope that in the present form the work and the correlation between this two important factors (i) the synthesis method and used stabilizer and (ii) the Mn doping will be more precisely, thorough and to the point discussed.

2.The results of photocatalytic experiments show the variation of activities of samples depends on Mn doping levels and synthetic processes. However, the origin of those variations can be hardly correlated to other characterization results in the current version of the manuscript. For instance, DR UV-vis spectra show the increase of the visible light absorption upon the Mn doping, but it is confusing how the increase of visible light absorption can correlate the photocatalytic activity under the UV light irradiations.

In fact, as can be seen in Table 1 and Figure 4a, the Mn doping led to a shift of the ZnS optical band gap from 3.30 to 2.72 eV. The Mn2+ ions doping caused also the enhanced absorbance to the visible region of the spectrum. However, the Mn doping in ZnS NCs allows them to be used as effectual UV photocatalysts for the degradation of organic contaminants. This enhancement can suggest a more efficient separation process of the photogenerated charge carriers than that of un-doped ZnS, which can positively influence the photocatalytic activity even under UV light. The Mn2+ ions in ZnS NCs behave like electron sinks, which can effectively trap and transfer the photogenerated electrons, and then improve the separation of electrons and hole pairs. Similar effect has been observed also for other nanomaterials e.g. ZnSe, ZnO, CuO by other authors [references 30, 35, 36 in manuscript].

Reviewer 2 Report

In the article, the authors describe a method to synthesize doped or undoped zinc sulfide nanocrystals via microwave and stabilize the nanocrystals with amine stabilizers. In addition, the nanocrystals are examined for photocatalytic application. The synthesis method is well introduced and results are clearly demonstrated. The topic fits Materials but there are some minor issues that need to be addressed.

Technical:

  1. In Table 1, could authors rearrange the ZnS_EN_Mn and ZnS_NH_Mn linearly? From 0.01 to 0.001 and then back to 0.02, is hard to follow. The same pattern is in the text as well. Line 203-205. Also in Figures 3, 4, 5. Please address.
  2. Figure 2, for figure h-j, the figure captions are missing.
  3. In Figure 3b, the undoped ZnS data are not presented while in the text, Line 237, it quotes “Further evidence for the surface properties of the ZnS and ZnS: Mn samples were obtained with FTIR spectroscopy.” Please address.
  4. Figure 4d, the legend is missing the red and olive lines. Also, what is the PL excitation wavelength?
  5. In Figure 5a, the ZnS_NH has absorption between 200 - 300 nm while in the figure, the absorption of Zn_NH_Mn is not clearly shown. Please describe the concentration ratio between ZnS_NH_Mn vs RhB.
  6. In the photodegradation experiment, could authors verify that the ZnS nanoparticles are or aren’t degraded during the processes?

Non-technical

  1. Line 269, “Vis” to visible, where Vis is not a standard phase outside UV-Vis.
  2. Figure 4, DC UV-Vis. DC is not expressed as full form anywhere.
  3. Line 136, 0.1 m KNO3, m should be capitalized.

Author Response

Response to Reviewer  2

We are very grateful to Reviewer for their comments and suggestions. The manuscript has been substantially reformatted and modified based on these suggestions. Detailed responses to each of the review’s comments are given below. Reviewer comments are in black and the responses to the reviewer comments are in blue, as well as the corrections in manuscript are in read. 

 Reviewer 2:

In the article, the authors describe a method to synthesize doped or undoped zinc sulfide nanocrystals via microwave and stabilize the nanocrystals with amine stabilizers. In addition, the nanocrystals are examined for photocatalytic application. The synthesis method is well introduced and results are clearly demonstrated. The topic fits Materials but there are some minor issues that need to be addressed.

Technical:

  1. In Table 1, could authors rearrange the ZnS_EN_Mn and ZnS_NH_Mn linearly? From 0.01 to 0.001 and then back to 0.02, is hard to follow. The same pattern is in the text as well. Line 203-205. Also in Figures 3, 4, 5. Please address.

All suggested changes was done in manuscript, corrections are in read (please see Tab. 1 and Figures 3,4, 5, line 223-229).

,, The results are presented in Table 1. In the case of the Mn-doped ZnSe NCs samples, the successful doping process of the manganese element in the crystal lattice of ZnS was observed with the loading percentage of about 0.0944, 0.1682, 0.4705, 3.744, 0.3706, 0.6847, 0,1207 and 0,3857 atomic % for ZnS_EN_Mn0.001, ZnS_EN_Mn0.005, ZnS_EN_Mn0.01, ZnS_EN_Mn0.02, ZnS_NH_Mn0.001, ZnS_NH_Mn0.01, ZnS_K_EN_Mn0.01 and ZnS_K_NH_Mn0.01, respectively.”

  1. Figure 2, for figure h-j, the figure captions are missing.

The  figure captions in fig. 2h-j were added (please see line 254-256 in manuscript).

3. In Figure 3b, the undoped ZnS data are not presented while in the text, Line 237, it quotes “Further evidence for the surface properties of the ZnS and ZnS: Mn samples were obtained with FTIR spectroscopy.” Please address.

In fact, the FT-IR undoped ZnS is not presented, the text in line 282 was modified be removing ‘’ZnS.’’

4. Figure 4d, the legend is missing the red and olive lines. Also, what is the PL excitation wavelength?

We would like to kindly apologize the Reviewer for the error. It was our mistake. In fig. 4d we presents only the photoluminescence (PL) excitation spectra (not wavelength) for samples ZnS_NH, ZnS_NH_Mn0.001, ZnS_NH_Mn0.01 and ZnS_K_NH_Mn0.01, all additional curves were removed.

PL is the abbreviation from photoluminescence spectroscopy. The explanation of the PL abbreviation was added in line 130.

5. In Figure 5a, the ZnS_NH has absorption between 200 - 300 nm while in the figure, the absorption of Zn_NH_Mn is not clearly shown. Please describe the concentration ratio between ZnS_NH_Mn vs RhB.

The concentration of dye RhB vs ZnS_NH_Mn photocatalyst  was 1:67 by weight, (typical 0.15 mg RhB vs 10 mg of ZnS:Mn NCs). The detailed  information was added in the section 2.5 and in legend of fig. 5a.

6. In the photodegradation experiment, could authors verify that the ZnS nanoparticles are or aren’t degraded during the processes?

The stability test of ZnS:Mn photocatalyst was conducted. Please see Figure 9 and text 489 – 498.  According to the result ZnS NCs are not degraded significantly in first 3 cycles, after that we observe some very small degradation process of nanocrystals.  

Non-technical

1. Line 269, “Vis” to visible, where Vis is not a standard phase outside UV-Vis.

The correction was done, line 309 in manuscript.

2. Figure 4, DC UV-Vis. DC is not expressed as full form anywhere.

Figure 4 presents diffuse reflectance ultraviolet-visible spectroscopy (DR UV-Vis). The full form of DR UV-Vis was added in line 128 in manuscript.

 3. Line 136, 0.1 m KNO3, m should be capitalized.

The correction was done, please see the line 147 in manuscript.

Reviewer 3 Report

The manuscript describes the synthesis of zinc sulfide and manganese-doped zinc sulfide nanostructures using a microwave-assisted solvothermal method. The addition of manganese shifted the light absorption of the synthesized ZnS material to the visible region. When ethylenediamine and hydrazine were used as stabilizers, one-dimensional ZnS nanowires with high specific surface area were obtained, regardless of whether manganese was present or not. The obtained materials were characterized by complementary methods and used for the degradation of rhodamine B with UV light (254 nm). The results presented are interesting, but before the manuscript can be recommended for publication, it needs to be thoroughly revised.

Questions and remarks

Was the degraded pollutant Rhodamine 6G or Rhodamine B. Both have a different CAS number and differ in their structure.

Are the ZnS nanowires single crystalline or did they compose of a big number of small crystallites (semi-crystalline)? The ZnS reflections can be used to estimate the crystallite size. Did the 2theta position of the ZnS reflections shifted at presence of manganese?

The authors should explain how from Figure 2 (h, i. j) the molar zinc to sulfur ratio of 1: 1 was derived.

The experimental determined atom% of manganese should be added to Table 1. The strong differences between nominal and experimental determined manganese content should be discussed in the text. What is the reason for the strong deviation.?

The BET surface area of ZnS_EN_Mn0.01 differs considerable from the BET surface area of the other samples. What is the reason for the high surface area? Was the high BET surface area of this sample confirmed by a repeated synthesis?

The method for calculating the optimal IPCE efficiency (0.1%) should be outlined in the manuscript. Incidentally, this value seems to be very low. To assess the photocatalytic performance of the synthesized ZnS/Mn materials, the degradation results should be compared with those of a reference catalyst such as P25 TiO2.

Is the method used to determine the band gap really accurate enough to distinguish between 3.78 and 3.81 eV? How large is the error with this method?

The materials have a relatively high BET surface. How many Rhodamine B was adsorbed during the dark experiment? This information should be added to the text.

Sample ZnS-NH_Mn0.01 showed the highest efficiency in the degradation process of all catalysts but its pseudo first order rate constant was clearly lower than those of some other catalysts. What is the reason for that?

The results of the stability tests are not clear. It would be better using only one catalyst (e.g. those with the highest activity) and repeat the experiment three or four times to see a clear tendency.

It is not necessary to repeat all the data presented in Table 1, Figure 6 and Figure 7 in the text of the manuscript.

Author Response

We are very grateful to Reviewer for their comments and suggestions. The manuscript has been substantially reformatted and modified based on these suggestions. Detailed responses to each of the review’s comments are given below. Reviewer comments are in black and the responses to the reviewer comments are in blue, as well as the corrections in manuscript are in read.

Response to Reviewer 3

The manuscript describes the synthesis of zinc sulfide and manganese-doped zinc sulfide  nanostructures using a microwave-assisted solvothermal method. The addition of manganese  shifted the light absorption of the synthesized ZnS material to the visible region. When  ethylenediamine and hydrazine were used as stabilizers, one-dimensional ZnS nanowires with  high specific surface area were obtained, regardless of whether manganese was present or not.  The obtained materials were characterized by complementary methods and used for the  degradation of rhodamine B with UV light (254 nm). The results presented are interesting, but  before the manuscript can be recommended for publication, it needs to be thoroughly revised.

Questions and remarks
Was the degraded pollutant Rhodamine 6G or Rhodamine B. Both have a different CAS number and differ in their structure.

Response
In fact, in the photocatalytic tests we used Rhodamine B with chemical formula: C₂₈H₃₁N₂O₃Cl, and CAS no. 81-88-9. The abbreviation of Rhodamine B and name of used dye was in whole manuscript again checked and unified. Additional the CAS number was added (please see line 85 in manuscript).

Are the ZnS nanowires single crystalline or did they compose of a big number of small crystallites (semi-crystalline)? The ZnS reflections can be used to estimate the crystallite size. Did the 2theta position of the ZnS reflections shifted at presence of manganese?

Response
The average crystallite size of nanomaterials was calculated using Scherrer’s formula. The average crystallite diameter of the all as-prepared ZnS and ZnS:Mn nanowires was about 5 ± 3 nm, which it can suggested that ZnS and ZnS:Mn NCs composed from a big number of small crystallites. Additionally, in fact, the diffraction peaks of (100), (002) and (101) planes of the Mn-doped ZnS NCs showed a slight shift towards the higher 2θ side compared with undoped 
ZnS. This peak position shift may occur due to the changes that occur in the lattice parameters of the host lattice on the incorporation of the dopant atoms.
According to review’s suggestion the discussion in manuscript was modified. Please see line 191 – 207 in manuscript.

The authors should explain how from Figure 2 (h, i. j) the molar zinc to sulfur ratio of 1: 1 was derived.
Response

In fact, it is not an accurate term used in this context. This is a kind of overinterpretation. From the fig. 2 h, i and j we cannot derived how exactly is the molar ration of zinc and sulfur and  manganese in ZnS:Mn NCs. Therefore we decide to modified this sentence as fallow: 
,,The corresponding EDS analysis (Figure 2 h, i, j; sample ZnS_EN_Mn0.01) confirms that ZnS:Mn NWs contain zinc, sulfur and Mn-doped elements evenly distributed in the crystal  lattice of ZnS.” (line 220 – 222).

The experimental determined atom% of manganese should be added to Table 1. The strong differences between nominal and experimental determined manganese content should be discussed in the text. What is the reason for the strong deviation.?

Response
The experimental determined atom% of manganese was added to Table 1. The strong differences between nominal and experimental determined manganese content in ZnS:Mn NCs is also discussed in manuscript. Please see the line 222-242. The mean reason of deviation is that thermal equilibrium, which requires facile diffusion in colloidal synthesis, may be far from realized.

The BET surface area of ZnS_EN_Mn0.01 differs considerable from the BET surface area of the other samples. What is the reason for the high surface area? Was the high BET surface area of this sample confirmed by a repeated synthesis?

Response
We would like to kindly apologize the Reviewer for the error, which appear in the paper and it concerning the sample named in the text “ZnS_EN_Mn0.01". Before the measurement accidentally we did not introduced the value of the mass of the sample. In that case apparatus using the mass of sample which were introduced at the beginning of the series of measurements. As proof for this fact we would like to enclose the original reports from the sorption experiments. The files were named as followed: AV00530.pdf (first file in the series) 
AV00536.pdf (file with incorrect mass of the sample) AV00536 corr.pdf (file with the corrected mass of the sample). This error was corrected in manuscript. The surface area of ZnS_EN_Mn0.01 is 207.4 m2/g. In table 1 we also added the average pore width of all samplesin Table 1. All information was discussed, see line 260 – 280. 

The method for calculating the optimal IPCE efficiency (0.1%) should be outlined in the manuscript. Incidentally, this value seems to be very low. To assess the photocatalytic performance of the synthesized ZnS/Mn materials, the degradation results should be compared with those of a reference catalyst such as P25 TiO2.

Response
We would like to kindly apologize the Reviewer for the error, as figure 8 we added not the incident photon-to-electron conversion efficiencies (IPCE) as a function of wavelength and potential for ZnS_NH_Mn0.01 but the photocurrent action map ( photocurrent values versus light wavelength and applied potential) for ZnS_NH_Mn0.01. In revised version of our manuscript the method for calculating the optimal IPCE and the IPCE map is added (Figure 8) for ZnS_NH_Mn0.01. The IPCE according to result is about 5%, not 0.1% - it was our 
mistake. The IPCE of P25 TiO2 according to literature is 57%. 

Is the method used to determine the band gap really accurate enough to distinguish between 3.78 and 3.81 eV? How large is the error with this method?

Response

According to literature (Sangiorgi, N.; Aversa, L.; Tatti, R.; Verucchi, R.; Sanson, A., 
Spectrophotometric method for optical band gap and electronic transitions determination of semiconductor materials. Opt. Mater., 2017, 64,18–25) we are convinced that it is possible to distinguish optical band gap between 3.78 and 3.81 eV for studied samples, but as the Reviewer rightly pointed out to us, we need to deliver the information about the error of this method. 
According to the literature spectrophotometric method is a rapid and relatively precise techniques for optical band gap determination for powder samples with the standard error about ± 0.02. This information was added in text, see line 317-318. In Table 1 the information about the error of the used method also was added. 

The materials have a relatively high BET surface. How many Rhodamine B was adsorbed during the dark experiment? This information should be added to the text.

Response
In fact, the BET surface is relatively high. We observed that about 15% of Rhodamine B is adsorbed during the dark experiment. This information was added in manuscript, please see line 360-363 in manuscript.

Sample ZnS-NH_Mn0.01 showed the highest efficiency in the degradation process of all catalysts but its pseudo first order rate constant was clearly lower than those of some other catalysts. What is the reason for that? 

Please see line 453-467.
,,In fact, ZnS_NH_Mn0.01 sample shows the highest efficiency in the degradation process of all catalysts but its pseudo first order rate constant was clearly lower than those of some other catalysts. The pseudo first order rate constant was 0.9496, 0.9434, 0.9402, 0.7842 and 0.9577 for the ZnS_En, ZnS_EN_Mn0.01, ZnS_NH and ZnS_NH_Mn0.01, ZnS_K_NH_Mn 0.01 samples, respectively, as shown in Figure 6. It can be found that the fitting of experimental data 
to the pseudo-first-order model was not so ideal, with low correlation coefficients of 0.7842 for ZnS_NH_Mn0.01 indicating that the experimental data did not well obey the pseudo-first-order kinetic model. It is known that photocatalytic oxidation rate constant, k, is property of the photocatalyst, and does not depend appreciably upon reactant structure [37]. Therefore this 
deviation can be connected to different aspects related with the surface properties of photocatalysis as well as the reaction mechanism including the adsorption/diffusion process; e.g. the photocatalyst surface was saturated in dye over nearly the entire concentration range studied. However, in order to give a clear explanation of this deviation the further research of the system, including an understanding of the adsorption mechanisms and the detailed kinetic 
analysis are required [38].”

The results of the stability tests are not clear. It would be better using only one catalyst (e.g. those with the highest activity) and repeat the experiment three or four times to see a clear tendency.

Response
According to Review’s suggestion the stability test was repeated for the best samplesZnS_NH_Mn 0.01. Please see Figure 9.

It is not necessary to repeat all the data presented in Table 1, Figure 6 and Figure 7 in the text of the manuscript

Reviewer 4 Report

Thank you very much for your interesting paper.
Since effects of stabilizer on photo-catalytic properties of the ZnS nanowires were experimentally investigated in detail in this study, this paper will be useful for researchers in this research fields.
However, there are some points should be revised in this paper, I would like to ask you to revise this paper with taking account of the following comments.
1) Explanations of the XRD peaks in Fig.1 are required.
2) The positions Fig. 2 h) and i) were taken seem to be different from those Fig.2 a) and b). Fig.2 h) and i) should be taken at the same position as those of Fig.2 a) and b).
3) In addition, there is no information on Mn in Fig.2 j). Please explain the reason why.
4) Please check the sentence “In turn, in the case of ………for ZnS_NH_Mn0.001.” again. I think “in the cases of” and “in comparison with” are better than “in the case of” and “when compared to”, respectively.
5) Please tell the reason why the data of ZnS_NH_Mn0.01 in Fig.6 a) were different from the linear line from 60 min. to 100 min. 
6) Please add the lines between plots to Fig. 9 b).

That is all.
I would very much appreciate your contribution.

Author Response

We are very grateful to Reviewer for their comments and suggestions. The manuscript has been substantially reformatted and modified based on these suggestions. Detailed responses to each of the review’s comments are given below. Reviewer comments are in black and the responses to the reviewer comments are in blue, as well as the corrections in manuscript are in read.

Response to Reviewer 4

Thank you very much for your interesting paper. Since effects of stabilizer on photo-catalytic properties of the ZnS nanowires were experimentally investigated in detail in this study, this paper will be useful for researchers in this research fields. However, there are some points should be revised in this paper, I would like to ask you to revise this paper with taking account of the following comments.

1) Explanations of the XRD peaks in Fig.1 are required.

According to Review’s suggestion in figure 1 all characteristic peaks have been described, please see Figure 1 and line 186-191.

,, The phases and the purities of the nanomaterials were investigated with the XRD analysis. Figures 1a and 1b indicate XRD patterns of ZnS and ZnS:Mn NCs. The strong and broad diffraction peaks appearing in the XRD diagrams at 27.0°, 28.6°, 30.5°, 39.6°, 47.9°, 52.0° and 56.6° have obvious relevance to the well-crystallized hexagonal wurtzite ZnS structures (W, JCPDS#96-110-0045) matching relatively well with the crystal planes of (100), (002), (101), (102), (110), (103), (112).”

2) The positions Fig. 2 h) and i) were taken seem to be different from those Fig.2 a) and b). Fig.2 h) and i) should be taken at the same position as those of Fig.2 a) and b). 3) In addition, there is no information on Mn in Fig.2 j). Please explain the reason why.

According to Review’s suggestion in figure 2 we added a new TEM micrographs in STEM-HAADF mode (see fig. 2h) and correlated EDX mapping for Zn, S and Mn elements. The quality of Mn element mapping image  (figure 2 k) now is much better than the image placed in the previous version of the manuscript. The distribution of Mn element in the test sample is now observed.  Taking into account the high cost of TEM analysis we decided to keep the TEM analyses for other samples that illustrate only their morphology.

4) Please check the sentence “In turn, in the case of ………for ZnS_NH_Mn0.001.” again. I think “in the cases of” and “in comparison with” are better than “in the case of” and “when compared to”, respectively.

The text in manuscript was modified according to Review’s suggestions, please see line 326-330.

,, In the case of ZnS and ZnS:Mn stabilized with hydrazine, we also observe slightly increased band gap values in comparison with the ZnS bulk value materials, i.e. 3.78 eV for ZnS_NH, 3.81 eV for ZnS_NH_Mn0.001.”

5) Please tell the reason why the data of ZnS_NH_Mn0.01 in Fig.6 a) were different from the linear line from 60 min. to 100 min. (Review 4)

and

 Sample ZnS-NH_Mn0.01 showed the highest efficiency in the degradation process of all catalysts but its pseudo first order rate constant was clearly lower than those of some other catalysts. What is the reason for that? (Review 3)

Please see line 453-467.

,,In fact, ZnS_NH_Mn0.01 sample shows the highest efficiency in the degradation process of all catalysts but its pseudo first order rate constant was clearly lower than those of some other catalysts. The pseudo first order rate constant was 0.9496, 0.9434, 0.9402, 0.7842 and 0.9577 for the ZnS_En, ZnS_EN_Mn0.01, ZnS_NH and ZnS_NH_Mn0.01, ZnS_K_NH_Mn 0.01 samples, respectively, as shown in Figure 6. It can be found that the fitting of experimental data to the pseudo-first-order model was not so ideal, with low correlation coefficients of 0.7842 for ZnS_NH_Mn0.01 indicating that the experimental data did not well obey the pseudo-first-order kinetic model. It is known that photocatalytic oxidation rate constant, k, is property of the photocatalyst, and does not depend appreciably upon reactant structure [37]. Therefore this deviation can be connected to different aspects related with the surface properties of photoca-talysis as well as the reaction mechanism including the adsorption/diffusion process; e.g. the photocatalyst surface was saturated in dye over nearly the entire concentration range studied. However, in order to give a clear explanation of this deviation the further research of the system, including an understanding of the adsorption mechanisms and the detailed kinetic analysis are required [38].”

6) Please add the lines between plots to Fig. 9 b).

According to all reviews (1-4) suggestion the stability test was modified.  Pleases see figure 9.

Round 2

Reviewer 1 Report

I believe that the overall quality and merit of the revised manuscript are still too low to be published in Materials.

1. Based on the authors’ reply to my previous comments, I assumed that authors want to discuss the impacts of stabilizing agents (EN and NH) and Mn doping on the photocatalytic activities of synthesized ZnS nanowires. Although authors separate the paragraphs 3.2 “Photocatalytic activity” into 3.2.1 and 3.2.2, there are still a lot of issues that need to be addressed.

- Overall, the results of photocatalytic activities of various samples were not correlated with the discussions of characterization results. Besides, characterization results of various samples should be more carefully examined and the way of presentation of datum needs to be improved (including data selection displayed in one graph for comparison).

[XRD patterns]

- Selection of stabilizer (EN or NH) resulted in different XRD patterns regardless of Mn doping, although XRD patterns of all the ZnS samples exhibited characteristic peaks of crystalline ZnS. The relative peak ratio of (002) to those of (100) and (001) decreased when NH was used as a stabilizer instead of EN. It is not mentioned or explained, either. It is worth mentioning that two non-Mn doped ZnS samples synthesized with EN and NH, respectively (denoted as ZnS_EN and ZnS_NH in this manuscript), did exhibit different photocatalytic activities towards RhB degradation under 265 nm irradiation. In addition, the other results indicated that two samples (ZnS_EN and ZnS_NH) were different in terms of light absorption at UV region, PL emission, surface area, and pore width. However, those were not compared and discussed properly.

- If ZnS was synthesized using the same stabilizer, variation of Mn doping did not seem to induce significant changes in XRD patterns, excepting for the slight 2theta shift of the entire XRD pattern of ZnS_NH upon the Mn doping (ZnS_NH vs ZnS_NH_Mn0.01 in Figure 1b)). It seemed that a similar 2 theta shift upon Mn doping was not observed in the case of the ZnS_EN sample (Figure 1a) (at least it is not clear). However, the authors mentioned that “However, the diffraction peaks of (100), (002) and (101) planes of the Mn-doped ZnS NCs showed a slight shift towards the higher 2theta side compared with undoped ZnS” (line 203-204) which can mislead the readers. Moreover, in the following lines (line 204-207), the authors attributed the 2 theta shift upon Mn doping which was only observed in the case of ZnS_NH to the incorporation of the Mn (Mn2+) having a different ion radius from Zn (Zn2+). This cannot explain why the ZnS_EN sample did not exhibit any noticeable peak shift upon the Mn doping. Even the ZnS_En_Mn0.02 sample with the highest amount of Mn doping (3.7447 at%) among all the samples studied in this work showed similar XRD patterns with the ZnS_En sample. In addition, there was also no shift of XRD peak positions of the ZnS_K_NH_Mn0.01 sample from those of ZnS_NH. Do those mean that it was only the ZnS_NH_Mn samples Mn2+ ions substituted the Zn2+ sites upon Mn doping? Those needs more careful examination of datum and explanations.

[the Mn at% in table 1]

- As the authors mentioned in lines 231-233, the exact doping contents of Mn in ZnS can differ from the nominal values which were determined based on the relative amount of Mn precursors used in the synthetic process. The results also indicated that the natures of stabilizers can result in different effectiveness of the Mn doping in ZnS. NH seemed to be more effective for Mn doping in ZnS compared to EN; 0.4705 Mn at% of ZnS_EN_Mn0.01 vs 0.6847 Mn at% of ZnS_NH_Mn0.01. In addition, MW was more effective than conventional heating regardless of the types of stabilizers. The related discussions can be also included in this work.

- The comparison of characterization and photocatalytic experiment results among samples must be made considering the nominal Mn doping values were different from the exact doping amount of Mn. For instance, ZnS_EN_Mn0.01 (0.4705 of Mn at%) has different doping amount from ZnS_NH_Mn0.01 (0.6847 of Mn at%), ZnS_K_EN_Mn0.01 (0.1207 of Mn at%), and ZnS_K_NH_Mn0.01 (0.3857 of Mn at%). However, those were not considered when the author made a comparison among the samples to discuss the impacts of Mn doping on their photocatalytic activities and characterization results.

[the TEM images, Figure 2]

- It is better to present TEM images in the same magnification (at least placing the same scale bar), especially for those authors who want to make direct comparisons (e.g., figure 2a vs 2c vs 2e,f,g, and figure 2b vs 2d). It seemed that ZnS_EN_Mn0.01 exhibited a similar structure with ZnS_K_EN_Mn0.01 if one compared figure 2h and figure 2e in similar magnification (although figure 2h is STEM image whereas figure 2e is TEM image). I wonder what the TEM image of ZnS_EN_Mn0.01 looks if it is in the same magnification as the TEM image of ZnS_K_EN_Mn0.01 (figure 2e).

- One may notice differences in structures by comparing the images of various samples, such as ZnS_En_Mn0.01 (Figure 2a), ZnS_NH_Mn0.01 (Figure 2c), ZnS_K_NH_Mn0.01 (Figure 2e), ZnS_K_NH_Mn0.01 (Figure 2 f and g). However, those differences can be hardly attributed to a particular factor among the types of stabilizer, heating methods (MW or conventional), and doping levels of Mn, since those samples were even different in exact Mn doping levels as shown in table 1.

- Line 248-251 reads “In the case of ZnS:Mn stabilized with EN, the products is an agglomeration of one-dimensional nanowires with a longer length. In turn, the morphology of ZnS:Mn NCs stabilized with hydrazine is less regular than the NCs stabilized with EN and is the mixture of 1D and 2D NCs”. It is not clear whether the authors made a comparison between TEM images of ZnS_K samples (conventional heating) synthesized with EN and NH or those of ZnS samples synthesized with EN_NH (MW).

[the isotherms of N2 adsorption/desorption, Figure 3a]

- There are so many isotherms displayed in one graph as figure 3a, which made it very difficult to check the shapes of isotherms. It seemed that adsorption and desorption curves were almost overlapped in the case of some samples, but it is difficult to see it. Relating to the isotherm curves, the authors only mentioned that all the samples exhibited a typical type-IV isotherm with a distinct hysteretic loop.

- In lines 268-269, the authors concluded that “the key factor influencing the size of the surface is the selection of heating method by comparing the four Mn-doped ZnS samples having same nominal Mn doping values. The exact doping levels of Mn were different among those four samples as already mentioned earlier in this letter, and results summarized in table 1 clearly implied that the exact doping levels of Mn can affect the surface area value (comparing surface area values of ZnS_En or ZnS_NH samples with different Mn at%). Therefore, authors need to select samples prepared with two heating methods (MW and conventional heating methods) having similar actual Mn doping levels to make such a conclusion.

2.

[the UV DRS datum, Figure 4a, b, and c]

- Since the photocatalytic activities of various samples were examined under UV light irradiation (254 nm), it is important to analyze/compare the light absorption abilities of samples in the UV light region (close to 254 nm) not in the visible light region. Apparently, light absorption spectra in the UV light region varied among samples depending on types of stabilizers, Mn doping levels, and heating methods (MW or conventional heating). However, those were not compared and discussed at all in the manuscript. For instance, the UV light absorption ability of ZnS_NH and ZnS_EN cannot be compared with figures 4 a and b since they were plotted as two separated graphs with Y-axes in arbitrary units.

- The optical band gaps of various samples were determined in this manuscript and those were summarized in table 1. In the case of ZnS_NH, bandgap values decreased as the Mn doping level increased, whereas the changes of band gap values upon the Mn doping levels were more complicated in the case of ZnS_EN; initially, the bandgap decreased with an increase of Mn doping levels, but it decreased and again decreased as the Mn doping level kept increasing. Those were not mentioned and discussed at all. If authors want to compare the Mn doping effects on the optical properties of two ZnS samples prepared with two different stabilizers, authors need to select samples having similar actual Mn doping levels (Mn at%) not nominal Mn doping values.

- The changes of optical band gap upon the nature of stabilizer (EN or NH) were more significant than those induced by Mn doping (it can be clear if one compares ZnS_EN with ZnS_NH). And the selection of different stabilizers did not induce a significant size difference in synthesized ZnS samples if MW was used instead of conventional heating as implied by TEM analysis results. Those were not correlated with the discussions in lines 322-325 and 330-334 explaining different bandgap values among various samples.

- Authors made a long discussion on the optical band gap values determined by UV-DRS results of various samples (making discussion on the absorption edge extension to the visible light upon Mn doping). However, those were not so important to explain the different photocatalytic activities of various samples since activities were determined under UV light irradiation 254 nm. The light absorption abilities of various samples in the UV light region would be more useful to explain the variation of photocatalytic activities among the samples.

[the PL emission spectra, Figure 4d, and e]

- The PL emission spectra would be also useful to explain the variation of photocatalytic activities among the samples since they can provide information on the electron-hole separation efficiency. However, first, the wavelength of excitation light should be provided. And second, emission spectra of ZnS_NH and ZnS_EN should be compared since they can help us to understand the impacts of stabilizers on optical properties and photocatalytic activities regardless of Mn doping. Now two spectra were plotted as two separated graphs with Y-axes in arbitrary units.

- Lastly, interpretation of PL peak intensity needs to be checked. During PL emission measurements, the sample surface was irradiated by light. Once light energy was absorbed by the sample, the electron was excited from VB to CB leaving a hole in VB. Then, the excited electron falls back to the hole in VB (electron-hole recombination) emitting energy as light (PL signal). Thus, usually higher the PL intensity from the sample indicated a higher probability of electron-hole recombination, i.e., lower electron-hole separation efficiency. Figures 4 d and e showed the higher PL intensity upon the higher Mn doping levels, however, authors seemed to try to relate the higher PL intensity to improved ZnS activities upon the higher Mn doping levels. Those need further explanations in detail if the authors were certain of it.

[the photocatalytic activity datum]

- Photocatalytic activities of ZnS_EN and ZnS_NH should be compared and discussed in detail with their characterization results since it can help us to understand the impacts of stabilizers on optical properties and photocatalytic activities regardless of Mn doping. Comparison of various samples with Mn doping needs to be more carefully made considering the difference between the nominal values of Mn doping levels and actual values of Mn doping levels.

Author Response

We are very grateful to Reviewer for your all very valuable comments and additional suggestions. The manuscript has been substantially reformatted and modified based on your all suggestions. Reviewer comments are in black and the responses to the reviewer comments are in blue, as well as the corrections in manuscript are also in blue.

Reviewer 3 Report

The authors answered all my questions. I recommend to publish the revised manuscript.

Author Response

We are very grateful to Reviewer for your all very valuable comments .

Reviewer 4 Report

Thank you very much for your revision.

Since this paper was revised properly along the reviewers' comments, this paper is thought to be publishable.

If possible, since there is no explanation on the definition of "degradation of dye (%)", I would like to ask you to add the explanation.

I would very much appreciate your contribution.

Author Response

Response to Reviewer 1

We are very grateful to Reviewer for your all very valuable comments and additional suggestions. The manuscript has been substantially reformatted and modified based on your all suggestions. Reviewer comments are in black and the responses to the reviewer comments are in blue, as well as the corrections in manuscript are also in blue.

Comments and Suggestions for Authors:

Thank you very much for your revision.

Since this paper was revised properly along the reviewers' comments, this paper is thought to be publishable.

If possible, since there is no explanation on the definition of "degradation of dye (%)", I would like to ask you to add the explanation.

I would very much appreciate your contribution.

Response:

According to the Reviewer's suggestion the definition of "degradation of dye (%) has been added. Please see lines 505-509.

“The % of dye degradation is calculated using the following equation:

% Degradation= (Ao - At/Ao) x100

where Ao is the absorbance at time t =0 min and At is the absorbance after time t min of treatment, Ao and At are recorded at λmax of dye (at 550 nm for RhB) [35].
